# Phosphorylated viral protein evades plant immunity through interfering the function of RNA-binding protein

Juan Li[1,2], Huimin Feng[2], Shuang Liu[1,2], Peng Liu[2], Xuan Chen[2], Jin Yang[2], Long He[2], Jian Yang ORCID[2]*, Jianping Chen[1,2]*

1 College of Agriculture and Biotechnology, Zhejiang University, Hangzhou, China, 2 State Key Laboratory for Managing Biotic and Chemical Threats to the Quality and Safety of Agro-products, Institute of Plant Virology, Ningbo University, Ningbo, China

* nather2008@163.com (JY); jianpingchen@nbu.edu.cn (JC)

**Data Availability Statement:** All relevant data are within the manuscript and its Supporting Information files.

## Abstract

Successful pathogen infection in plant depends on a proper interaction between the invading pathogen and its host. Post-translational modification (PTM) plays critical role(s) in plant-pathogen interaction. However, how PTM of viral protein regulates plant immunity remains poorly understood. Here, we found that S162 and S165 of Chinese wheat mosaic virus (CWMV) cysteine-rich protein (CRP) are phosphorylated by SAPK7 and play key roles in CWMV infection. Furthermore, the phosphorylation-mimic mutant of CRP (CRP^S162/165D) but not the non-phosphorylatable mutant of CRP (CRP^S162/165A) interacts with RNA-binding protein UBP1-associated protein 2C (TaUBA2C). Silencing of *TaUBA2C* expression in wheat plants enhanced CWMV infection. In contrast, overexpression of *TaUBA2C* in wheat plants inhibited CWMV infection. *TaUBA2C* inhibits CWMV infection through recruiting the pre-mRNA of *TaNPR1*, *TaPR1* and *TaRBOHD* to induce cell death and $H_2O_2$ production. This effect can be supressed by CRP^S162/165D through changing TaUBA2C chromatin-bound status and attenuating it's the RNA- or DNA-binding activities. Taken together, our findings provide new knowledge on how CRP phosphorylation affects CWMV infection as well as the arms race between virus and wheat plants.

## Author summary

Chinese wheat mosaic virus (CWMV) causes a damaging disease in cereal plants. However, CWMV interacts with host factors to facilitate virus infection is not clear yet. Here, we found that S162 and S165 of CWMV cysteine-rich protein (CRP) are phosphorylated by SAPK7 *in vivo* and *in vitro*. Mutational analyses have indicated that these two phosphorylation sites of CRP (CRP^S162/165D) promoting CWMV infection in plants, due to the supressed cell death and $H_2O_2$ production. Further investigations found the CRP^S162/165D can interact with TaUBA2C, while the non-phosphorylatable mutant of CRP (CRP^S162/165A) does not. Futhermore, we have determined that CRP^S162/165D and TaUBA2C interaction inhibited the formation of TaUBA2C speckles in nucleus to attenuate its RNA- and

**Funding:** This work was supported by: China Agriculture Research System from the Ministry of Agriculture of the P.R. China (CARS-03) for J.C., Ningbo Science and Technology Innovation 2025 Major Project, China (Q21C140013) for J.C. and K. C. Wong Magna Funding Ningbo University for J.Y. The funders had no role in study design, data collection and analysis, decision to publish, or preparation of the manuscript.

**Competing interests:** The authors have declared that no competing interests exist.

DNA-binding activity. We also showed that TaUBA2C recruit the pre-mRNA of *TaNPR1*, *TaPR1* and *TaRBOHD* to up-regulated these genes expressions and then induce cell death and $H_2O_2$ production in plant. This effect can be supressed by the expression of $CRP^{S162/165D}$, in a dose-dependent manner. Taken together, our discovery may provide a new sight for the arms race between virus and its host plants.

## Introduction

Post-translational modification (PTM) can greatly modify protein functions, including the functions involved in the responses against abiotic and/or biotic stresses [1,2]. Several PTMs have now been reported. These include protein phosphorylation [3–5], sumoylation [6], ubiquitination [7], glycosylation [8], N-Myristoylation [5], and S-acylation [9]. Among these PTMs, protein phosphorylation is considered as one of the most studied PTMs and plays vital roles in many cellular processes [10]. Protein phosphorylation is a reversible process, and the γ-phosphoryl group of ATP can covalently bind to the serine (S), threonine (T) or tyrosine (Y) residue in the target protein through the actions of protein kinases. Numerous studies have indicated that after phosphorylation, many viral proteins can enhance or suppress virus infection in plants. For instance, the phosphorylated *Barley stripe mosaic virus* (BSMV) γb protein has been shown to suppress RNA silencing and virus infection-induced cell death in plants [4]. *N. benthamiana* NbSKη can phosphorylate *Tomato leaf curl Yunnan virus* (TLCYnV) C4 protein to enhance virus pathogenicity [5,11]. The Casein Kinase 1 (CK1)-mediated phosphorylation of the serine-rich motif in *Barley yellow striate mosaic virus* (BYSMV) phosphoprotein is essential for viral replication [12]. However, how protein phosphorylation modulates virus pathogenicity in plants is still largely unknown.

During evolution, plants have evolved multiplex defense strategies against pathogen infections. For example, programmed cell death (PCD) can protect plants from pathogen invasion through eliminating damaged or pathogen-infected cells [13,14]. The most well-studied plant PCD is hypersensitive response (HR), which involves activations of PR genes and productions of reactive oxygen species and salicylic acid (SA) to eliminate invading pathogen by dead cells [15,16]. In the process of pathogen infection, a variety of defense-associated genes are induced in plants [17–19]. Regulation of gene expression by post-transcriptional modification in the context of defense is important for PCD and plant immunity [20]. Previous reports have also shown that plant RNA-binding proteins (RBPs) can regulate gene expressions through post-transcriptional modification [21,22]. Many RBPs contain an RNA recognition motif (RRM), also known as RNA binding domain (RBD), and a domain(s) that are required for protein-protein interactions [23]. Some RBPs have been demonstrated to regulate plant immune responses. For example, AtRBP-DR1 contains three RRMs and can positively regulate the SA-mediated signaling in *Arabidopsis* plant [24]. *Arabidopsis* Dicer-like 4 (AtDCL4) and Argonaute 2 (AtAGO2) have also been shown to modulate *Arabidopsis* defense responses to pathogen invasions [25,26]. To counteract plant defense, pathogens have also evolved to encode different proteins to evade host immunity. For instance, HopU1 of *Pseudomonas syringae* can disrupt the binding between baterial RNA and GRP7, a Glycine-rich RNA binding protein, to regulate the expression of a plant immune receptor [27]. In this study, we investigated how CWMV evades wheat and *N. benthamiana* defense response to enhance virus infection.

*Chinese wheat mosaic virus* (CWMV), a member in the genus *Furovirus*, family *Virgaviridae*, is an RNA virus with two positive-sense single-stranded RNAs (e.g., RNA1 and RNA2). In China, CWMV often infects its host plants, together with *Wheat yellow mosaic virus*

(WYMV), to cause severe disease symptoms and yield losses [28,29]. In the laboratory, CWMV can infect the model plant *Nicotiana benthamiana* through mechanical inoculation, which has been used as a very common model system for investigating the interaction between CWMV and plants [30]. CWMV RNA1 encodes three proteins: the replication-associated protein, RNA-dependent RNA polymerase (RdRp), and a movement protein. CWMV RNA2 encodes four proteins: a major capsid protein (CP), two minor CP-related proteins (e.g., N-CP and CP-RT), and a cysteine-rich protein (CRP)[28,31]. The CRP is an RNA silencing suppressor [32]. Many virus-encoded suppressors of RNA silencing (VSRs) are multifunctional proteins. For example, potyvirus helper component proteinases (HCPro) are not only VSRs, but also important for virus plant-to-plant transmission and viral polyprotein maturation [33]. *Cucumber mosaic virus* (CMV) 2b protein is also a VSR and involved in CMV systemic infection [34]. *Barley stripe mosaic virus* (BSMV) γb protein can suppress RNA silencing and promote RNA duplex unwinding and chloroplast-associated virus replication as well as cell-to-cell movement. In addition, after phosphorylation, BSMV γb protein can inhibit cell death to benefit virus infection [4,35,36]. Although CWMV CRP has been reported as a VSR, how it evades wheat resistance to benefit virus infection is unclear.

Here, we present evidence that CWMV CRP is phosphorylated at S162 and S165 by a serine/threonine-protein kinase SAPK7 *in vivo* and *in vitro*. Using a CWMV mutant encoding a CRP no longer phosphorylatable at S162 and S165 (e.g., CWMV[S162/165A]), the inoculated *N. benthamiana* and wheat plants did not show clear mosaic symptoms compared with CWMV wild-type (WT) infection, due mainly to cell death and $H_2O_2$ production. In contrast, using a phosphorylation mimics (e.g., CWMV[S162/165D]), the inoculated plants developed stronger disease symptoms than the WT CWMV-inoculated plants, due mainly to the supressed cell death and $H_2O_2$ production. In this study, we have also identified a CWMV infection inducible *RNA-binding protein* gene (*TaUBA2C*) in wheat that interacts with CRP[S162/165D], but not CRP[S162/165A]. Further analyses showed that silencing of *TaUBA2C* expression in wheat promoted CWMV infection, while overexpression of *TaUBA2C* in wheat inhibited CWMV infection. It is noteworthy that TaUBA2C can induce cell death and $H_2O_2$ production in wheat plants through up-regulation of *TaNPR1*, *TaPR1* and *TaRBOHD* expressions. Also, the TaUBA2C-induced cell death and $H_2O_2$ production can be supressed by the expression of CRP[S162/165D], in a dose-dependent manner. Futhermore, we have determined that CRP[S162/165D] can bind to TaUBA2C to interfear the formation of TaUBA2C speckles in nuclei to reduce its RNA- and DNA-binding activity. In summary, through this study, we have determined that phosphorylation of CRP promotes CWMV infection in plants. The results described in this paper provide new knowledge of the arms race between CWMV and its host plants.

## Results

### CWMV CRP can be phosphorylated in wheat

To investigate whether CWMV CRP can be phosphorylated in plant, plasmid expressing CRP-GFP fusion was introduced into wheat protoplasts via PEG 4000-based method. After overnight incubation at room temperature, total protein was extracted from the inoculated protoplasts and CRP-GFP enriched by immunoprecipitation. Comparative western blot analysis using GFP and phosphoserine-specific antibodies revealed in protoplasts expressing the fusion protein a 45 kDa band indicative for phosphorylated CRP-GFP. GFP expressing samples were processed in parallel as a control (Fig 1A). To identify the phosphorylation sites in CRP, the GFP-tagged protein was purified by affinity chromatography from total protein extracts of expressing wheat protoplast (Fig 1B, upper panel). The purified CRP-GFP was then analyzed using Q-Exactive liquid chromatography-tandem mass spectrometry (LC-MS/MS)

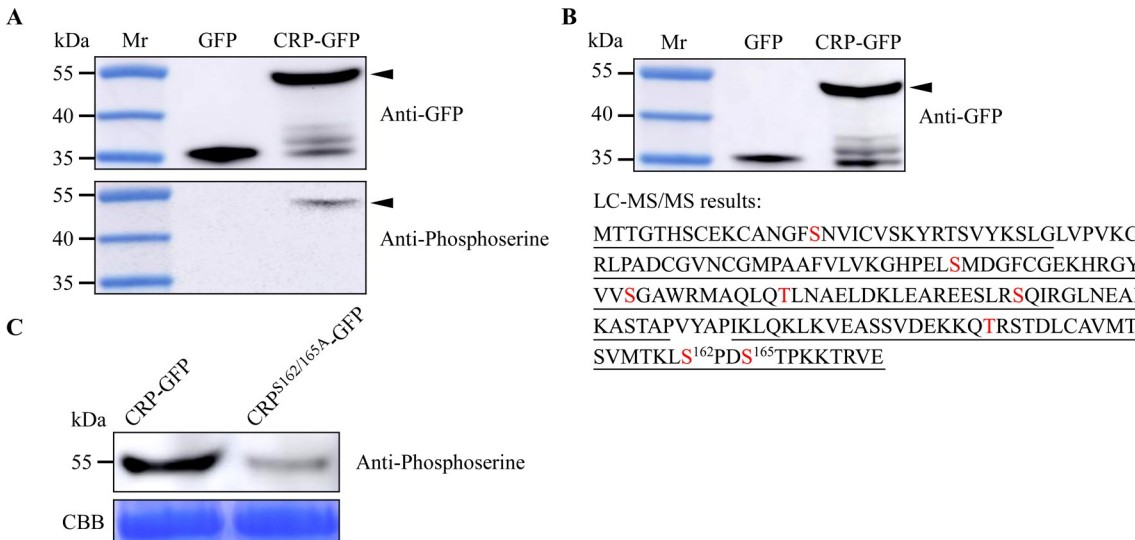

**Fig 1. Identification of phosphorylation sites in CWMV CRP through LC-MS/MS. A.** *In vivo* phosphorylation of CRP. Total protein was extracted from the protoplasts expressing GFP or CRP-GFP, and then analyzed through immunoprecipitation using an anti-GFP or an anti-phosphoserine antibody. **B.** Detection of CRP-GFP purified from the inoculated protoplasts using western blot (upper panel). The LC-MS/MS result is shown in the lower panel. The underlined CRP amino acid sequence was identified in this study through LC-MS/MS, and the phosphorylation sites in this protein are shown in red. **C.** The phosphorylation levels of CRP-GFP and CRP$^{S162/165A}$-GFP were determined through western blot analysis using a phosphoserine specific antibody. The CBB-stained gel is used to show sample loadings.

after trypsin treatment. Through 9 independent LC-MS/MS, a total of 8 potential phosphorylation sites were obtained (Fig 1B, lower panel). Considering, S162 and S165 were identified in 6 independent assays (S1 Table), these two sites were chosen for further assays. As previously described by Zhang *et al.*[4] and Gao *et al.*[12] with minor modifications, we substituted the two potential phosphorylation sites with alanine in plasmid expressing CRP-GFP fusion to investigate the phosphorylation levels of the WT and S162/165A versions of CRP in plant cells. We transiently expressed CRP$^{S162/165A}$-GFP in *N. benthamiana* leaves through agroinfiltration. The leaves expressing CRP-GFP were used as controls. Total protein was extracted from the inoculated leaves, then CRP-GFP and CRP$^{S162/165A}$-GFP enriched by immunoprecipitation. Western blot analysis showed that the phosphorylation level of CRP$^{S162/165A}$-GFP was significantly reduced (Fig 1C). Taken together, our results suggest that S162 and S165 phosphorylation sites are important.

## Phosphorylation of CRP at S162 or S165 promoted CWMV infection

In order to investigate the role of phosphorylation at S162 and/or S165 in CWMV infection, we generated six mutant CWMV infectious clones (CWMV$^{S162A}$, CWMV$^{S165A}$, CWMV$^{S162/165A}$, CWMV$^{S162D}$, CWMV$^{S165D}$, and CWMV$^{S162/165D}$), through substituting the serine residue (S) at the position 162 and/or 165 with an alanine to mimic the non-phosphorylated state, or with a aspartic acid to mimic the phosphorylated state. The WT and mutant infectious clones were inoculated individually to *N. benthamiana* plants through agroinfiltration. After 21 days, the CWMV$^{S162A}$- or CWMV$^{S165A}$-inoculated plants showed mild mosaic symptoms (Fig 2A). The CWMV$^{S162D}$- or CWMV$^{S165D}$-inoculated plants showed moderate mosaic symptoms, while the CWMV$^{S162/165D}$-inoculated plants showed strong mosaic symptoms. In this study, the CWMV$^{S162/165A}$-inoculated plants did not show clear mosaic symptoms (Fig 2A). Quantitative RT-PCR (qRT-PCR) result showed that the expression level of CWMV *CP* in the systemic leaves

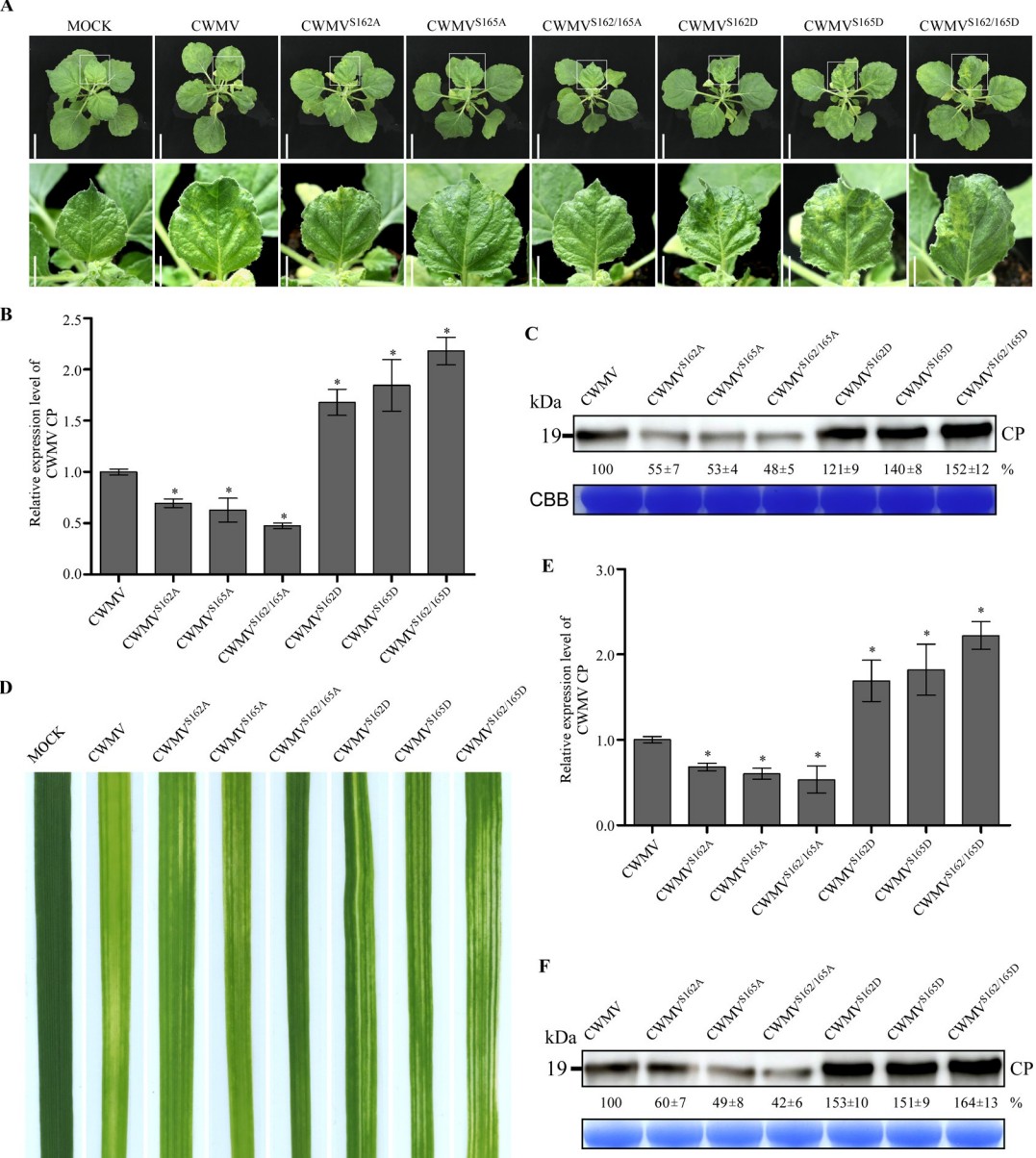

**Fig 2. Phosphorylation of CRP at S162 or S165 is crucial for CWMV infection. A.** Systemic mosaic symptoms in the CWMV-, CWMV[S162A]-, CWMV[S165A]-, CWMV[S162/165A]-, CWMV[S162D]-, CWMV[S165D]- or CWMV[S162/165D]-infected *N. benthamiana* plants. Photographs were taken at 21 dpi. Scale bar = 5 cm (upper panel), Scale bar = 2 cm (lower panel). **B.** Relative expression level of CWMV CP in the assayed *N. benthamiana* plants, determined through qRT-PCR using CWMV CP gene specific primers. The data presented are the means ± standard deviations (SD), calculated using the Student's *t*-test. Each treatment had three biological replicates. *, $P < 0.05$. **C.** Accumulation of CWMV CP in the assayed *N. benthamiana* leaf samples was determined through western blot analysis using a CWMV CP specific antibody. The CBB-stained gel is used to show sample loadings. **D.** Systemic mosaic symptoms in wheat leaves infected with CWMV, CWMV[S162A], CWMV[S165A], CWMV[S162/165A], CWMV[S162D], CWMV[S165D] or CWMV[S162/165D]. Photographs were taken at 21 dpi. **E.** The relative expression level of CWMV CP in the assayed wheat plants was determined through qRT-PCR using CWMV CP gene specific primers. The data presented are the means ± SD, determined using the Student's *t*-test. Each treatment had three biological replicates. *, $P < 0.05$. **F.** Accumulation of CWMV CP in the assayed wheat plants was determined through western blot analysis using a CWMV CP specific antibody. The CBB-stained gel is used to show sample loadings.

of the CWMV$^{S162A}$-, CWMV$^{S165A}$- or CWMV$^{S162/165A}$-inoculated plants were significantly reduced, while the expression level of CWMV *CP* in the systemic leaves of the CWMV$^{S162D}$-, CWMV$^{S165D}$- or CWMV$^{S162/165D}$-inoculated plants were significantly increased compared to the WT CWMV-inoculated plants (Fig 2B). Similar result was also obtained through western blot analysis using a CWMV CP specific antibody (Fig 2C).

We then inoculated wheat seedlings with the WT or mutant CWMV. By 21 dpi, the plants inoculated with CWMV$^{S162/165D}$ showed strong mosaic symptoms in their young developing leaves. The plants inoculated with CWMV$^{S162D}$ or CWMV$^{S165D}$ showed moderate mosaic symptoms, while the plants inoculated with CWMV$^{S162A}$ or CWMV$^{S165A}$ showed mild mosaic symptoms (Fig 2D). In this study, the plants inoculated with CWMV$^{S162/165A}$ again did not show clear mosaic symptoms in their leaves. The qRT-PCR and western blot results revealed that the accumulation levels of CWMV *CP* and CP were much higher in the plants inoculated with CWMV$^{S162D}$, CWMV$^{S165D}$ or CWMV$^{S162/165D}$ compared to the WT CWMV-inoculated plants (Fig 2E and 2F). In contrast, the plants inoculated with CWMV$^{S162A}$, CWMV$^{S165A}$ or CWMV$^{S162/165A}$ accumulated much less CWMV *CP* compared to the WT CWMV-inoculated plants, indicating that phosphorylation of CRP at S162 and/or S165 can promote CWMV infection in plant.

To investigate the effect of phosphorylation of CRP at S162 and S165 on its VSR activity, we expressed CRP, CRP$^{S162/165A}$ and CRP$^{S162/165D}$ in 16c transgenic *N. benthamiana* leaves as reported [4,37]. By 3 or 6 dpi, the green fluorescence in the 16c plant leaves expressing sGFP was silenced (S1A Fig). Consistent with this result, no GFP protein was detected in these GFP-silenced 16c plant leaves through western blot analysis (S1B Fig). However, the 16c plant leaves co-expressing sGFP and TBSV p19, sGFP and CRP, sGFP and CRP$^{S162/165A}$ or sGFP and CRP$^{S162/165D}$ continued to show strong green fluorescence (S1A Fig). This finding was supported by the result of western blot analysis (S1B Fig), suggesting that phosphorylation of CRP at S162 and S165 does not affect its RNA silencing suppression activity.

## SAPK7 kinase is responsible for the phosphorylation at S162 or S165

To identify which kinase(s) phosphorylates CRP at S162 or S165, we screened a wheat cDNA library through yeast two hybrid (YTH) assays using CRP as the bait. Through three independent screenings, a total of 15 positive clones were obtained (S2 Table). After sequence analysis, six of these clones were found to encode a polypeptide of 355 amino acids (aa). Blast search result showed that this polypeptide shared 95.8% sequence identity with the serine/threonine-protein kinase SAPK7 (TraesCS2A02G303900.1, named as TaSAPK7 thereafter), and this polypeptide represents the full-length of TaSAPK7. Further analysis revealed that wheat has three *TaSAPK7* homology sequences (TraesCS2A02G303900.1, TraesCS2B02G320500.1, and TraesCS2D02G302500.1), and these three sequences share 99.91% sequence identity (S2A Fig). In addition, phylogenetic analysis showed that SAPK7 is a member of the SnRK2 subfamily of plant protein kinase (S2B Fig). For convenience, we selected TraesCS2A02G303900.1 for our further analysis. To validate the function of this protein, we constructed a pAD-TaSAPK7 plasmid and used it in further YTH assays. The result showed that AD-TaSAPK7 interacted with BD-CRP in the yeast cells (Fig 3A). To confirm this finding, we conducted bimolecular fluorescence complementation (BiFC) and firefly luciferase (LUC) complementation imaging assays, and confirmed that TaSAPK7 interacted with CRP (Fig 3B and 3C).

To investigate whether TaSAPK7 phosphorylates CRP, we purified GST-TaSAPK7 and CRP-6×His from *E. coli*, and subjected them to *in vitro* kinase assay in 1× kinase reaction buffer supplemented with [γ-$^{32}$P] ATP. The result showed that TaSAPK7 was responsible for phosphorylating CRP (Fig 3D). In this study, we have also found that phosphorylation of CRP

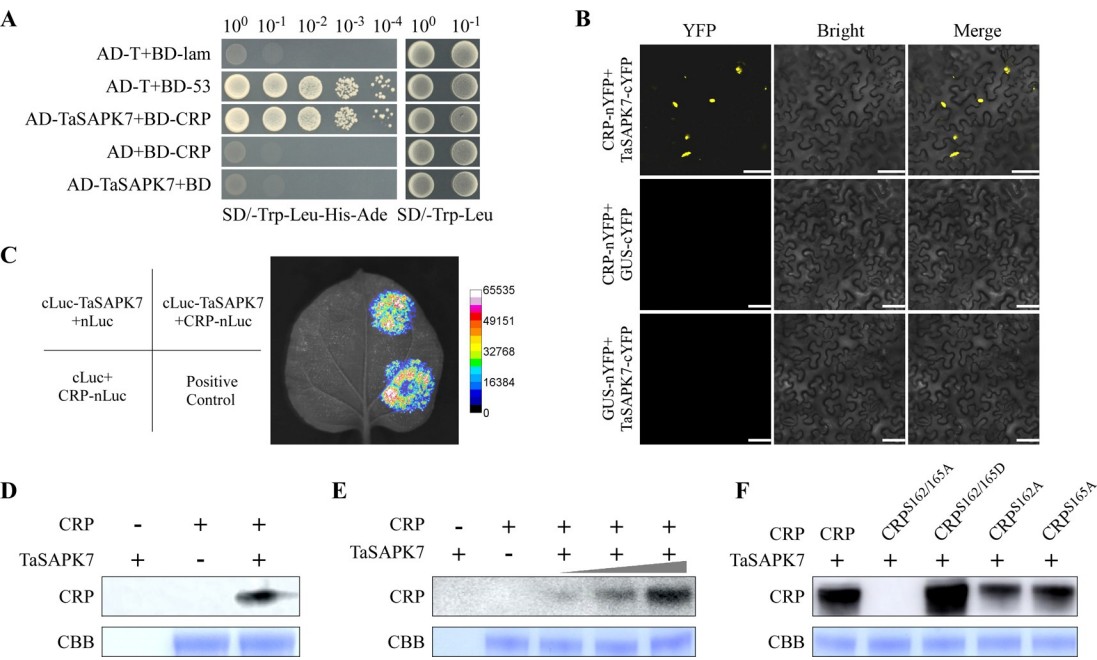

**Fig 3. CWMV CRP can be phosphorylated by TaSAPK7. A.** Yeast two-hybrid (YTH) assay was first used to determine the interaction between TaSAPK7 and CRP (TaSAPK7+CRP). AD-TaSAPK7 and BD-CRP were co-expressed in yeast cells, and the transformed cells were grown on the SD/-Leu/-Trp medium and then on the SD/-Trp/-Leu/-His/-Ade medium to determine the protein-protein interaction. Yeast cells co-transformed with AD-T+BD-Lam, AD+BD-CRP and AD- TaSAPK7+BD were used as the negative controls, yeast cells co-transformed with AD-T+BD-53 were used as the positive controls. **B.** BiFC assay was used to confirm the interaction between TaSAPK7-cYFP and CRP-nYFP in *N. benthamiana* leaves. These two proteins were co-expressed in *N. benthamiana* leaves followed by Confocal Microscopy at 60 h post agroinfiltration (hpi). Scale bar = 20 μm. **C.** LCI assay was used to confirm the interaction between cLuc-TaSAPK7 and CRP-nLuc in *N. benthamiana* leaves. Agrobacterium cultures carrying pCRP-nLuc or pcLuc-TaSAPK7 were mixed (cLuc-TaSAPK7+CRP-nLuc), and infiltrated into *N. benthamiana* leaves. The leaf areas infiltrated with Agrobacterium cultures expressing sGF-nLuc+cLuc-SAR were used as the positive controls, CRP-nLuc+cLuc and nLuc+cLuc-TaSRK7 were used as the negative controls, respectively. Luciferase activity was captured using a low-light cooled CCD imaging apparatus at 3dpi. **D.** Detection of kinase activity after CRP and TaSAPK7 interaction *in vitro*. In this study, recombinant CRP-6×His and TaSAPK7-GST were separately expressed and purified from *E. coli*, and then incubated together. The reactions lacking CRP-6×His or TaSAPK7-GST were used as controls. The CBB-stained gel is used to show sample loadings. **E.** Detection of the effect of TaSAPK7 on CRP phosphorylation *in vitro*. The radioactive protein bands indicate the levels of CRP phosphorylation. The CBB-stained gel is used to show sample loadings. **F.** Analysis of phosphorylations of CRP and its mutants by TaSAPK7. The radioactive protein bands indicate the levels of phosphorylation of various proteins. The CBB-stained gel is used to show sample loadings.

was TaSAPK7 dose-dependent (Fig 3E). Because CRP has several consensus phosphorylation sites, we substituted S162 and/or S165 with A or D, respectively, and performed further *in vitro* phosphorylation assays. The result showed that CRP[S162A] and CRP[S165A] were phosphorylated by TaSAPK7, but not CRP[S162/165A] (Fig 3F), further confirming that SAPK7 kinase is responsible for the phosphorylation of CRP at S162 and S165. However, we unexpectedly observed that the CRP[S162/165D] mutant showed a strong phosphorylation level (Fig 3F), but aspartic acid should not be phosphorylated by TaSAPK7. It suggested that either some sites other than S162 and S165 could be phosphorylated by TaSAPK7 and their phosphorylation is dependent on the phosphorylation of S162 and/or S165 or their Asp substitutes, or the Ser-to-Asp mutations resulted in new phosphorylation sites for TaSPAK7. To clarify the secondary phosphorylation sites of CRP, we analyzed the phosphorylation status of CRP[S162/165D] by LC-MS/MS. The results showed that S156 and T159 are secondary phosphorylation sites of CRP (S3A Fig). To investigate the role of phosphorylation at S156 and T159 in CWMV infection, we generated two mutant CWMV infectious clones CWMV[S156/T159A] to mimic the non-

phosphorylated state, and CWMV$^{S156/T159D}$ to mimic the phosphorylated state. The WT and mutant infectious clones were inoculated individually to *N. benthamiana* plants through agroinfiltration. After 21 days, the CWMV$^{S156/T159A}$- or CWMV$^{S156/T159D}$-inoculated plants showed mosaic symptoms similar to WT CWMV (S3B Fig). The qRT-PCR and western blot results revealed that the accumulation levels of CWMV *CP* and CP in plants inoculated with CWMV$^{S156/T159A}$ or CWMV$^{S156/T159D}$ were similar to those in plants inoculated with WT CWMV (S3C and S3D Fig). These results suggested that phosphorylation of CRP at S156 and T159 does not affect the pathogenicity of CWMV.

## SRK promotes CWMV infection

Considering CWMV can infect the model plant *Nicotiana benthamiana*, which has been used as a very common model system for investigating the interaction between virus and plants. Firstly, we search the homologous gene of *TaSAPK7* through NCBI database. The blast search result showed that TaSAPK7 have 80.39% amino acid identity to the serine/threonine-protein kinase SRK2A-like (LOC107817827, named as NbSRK thereafter) (S2C Fig). Through YTH assays, we have found that AD-NbSRK could also interact with BD-CRP (Fig 4A). This finding was then validated through BiFC and LCI assays (Fig 4B and 4C). Next, in order to investigate the role of SRK in CWMV infection, we first silenced *NbSRK* expression in *N. benthamiana* plants using a TRV-based VIGS technology. qRT-PCR result showed that the expression level of *NbSRK* was knocked down by about 70% in the TRV:NbSRK-infected plants compared to the TRV:00-infected plants (S4A Fig). These plants were then inoculated again with CWMV. At 21 days post CWMV inoculation, the TRV:NbSRK- and CWMV-inoculated (referred to as TRV:NbSRK +CWMV thereafter) plants showed milder mosaic symptoms than the plants inoculated with TRV:00 and CWMV (TRV:00+CWMV) (Fig 4D). qRT-PCR result showed that the accumulation level of CWMV CP in the TRV:NbSRK+CWMV-inoculated plants was decreased by 0.5 fold compared to the TRV:00+CWMV-inoculated plants (Fig 4E). Western blot result showed that the accumulation level of CWMV CP in the TRV:NbSRK+CWMV-inoculated plants was reduced by about 50% compared to the TRV:00+CWMV-inoculated plants (Fig 4F). To investigate that the effect of CRP phosphorylation on CWMV infection, we inoculated the NbSRK-silenced plants with CWMV$^{S162/165A}$ (TRV:NbSRK+CWMV$^{S162/165A}$) or CWMV$^{S162/165D}$ (TRV: NbSRK+CWMV$^{S162/165D}$). By 21 dpi, the TRV:NbSRK+CWMV$^{S162/165A}$-inoculated plants showed the similar mosaic symptoms as the TRV:NbSRK+CWMV-inoculated plants, while the TRV:NbSRK+CWMV$^{S162/165D}$-inoculated plants showed stronger mosaic symptoms than the TRV:NbSRK+CWMV-inoculated plants (Fig 4D). qRT-PCR result showed that the accumulation level of CWMV CP in the TRV:NbSRK+CWMV$^{S162/165A}$-inoculated plants was similar to that in the TRV:NbSRK+CWMV-inoculated plants. In contrast, the accumulation level of CWMV CP in the TRV:NbSRK+CWMV$^{S162/165D}$-inoculated plants were about 2.2 fold higher than that in the TRV:NbSRK+CWMV-inoculated plants (Fig 4G). Western blot result agreed with the qRT-PCR result and showed that the infection of CWMV$^{S162/165D}$, but not CWMV$^{S162/165A}$, in the NbSRK-silenced plants was enhanced (Fig 4H).

To further elucidate the role of NbSRK in CWMV infection, we constructed a p35S: NbSRK-His expression vector and co-inoculated this vector with CWMV (NbSRK+CWMV), CWMV$^{S162/165A}$ (NbSRK+CWMV$^{S162/165A}$) or CWMV$^{S162/165D}$ (NbSRK+CWMV$^{S162/165D}$) into *N. benthamiana* leaves through agroinfiltration. The plants co-inoculated with the empty vector and CWMV (EV+CWMV) were used as controls. At 3 dpi, the inoculated leaves were harvested and analyzed for NbSRK-His expression through western blot assays (S4B Fig). These plants were then analyzed through qRT-PCR at 7 dpi and the result showed that the accumulation level of CWMV CP was significantly increased in the NbSRK+CWMV- or the

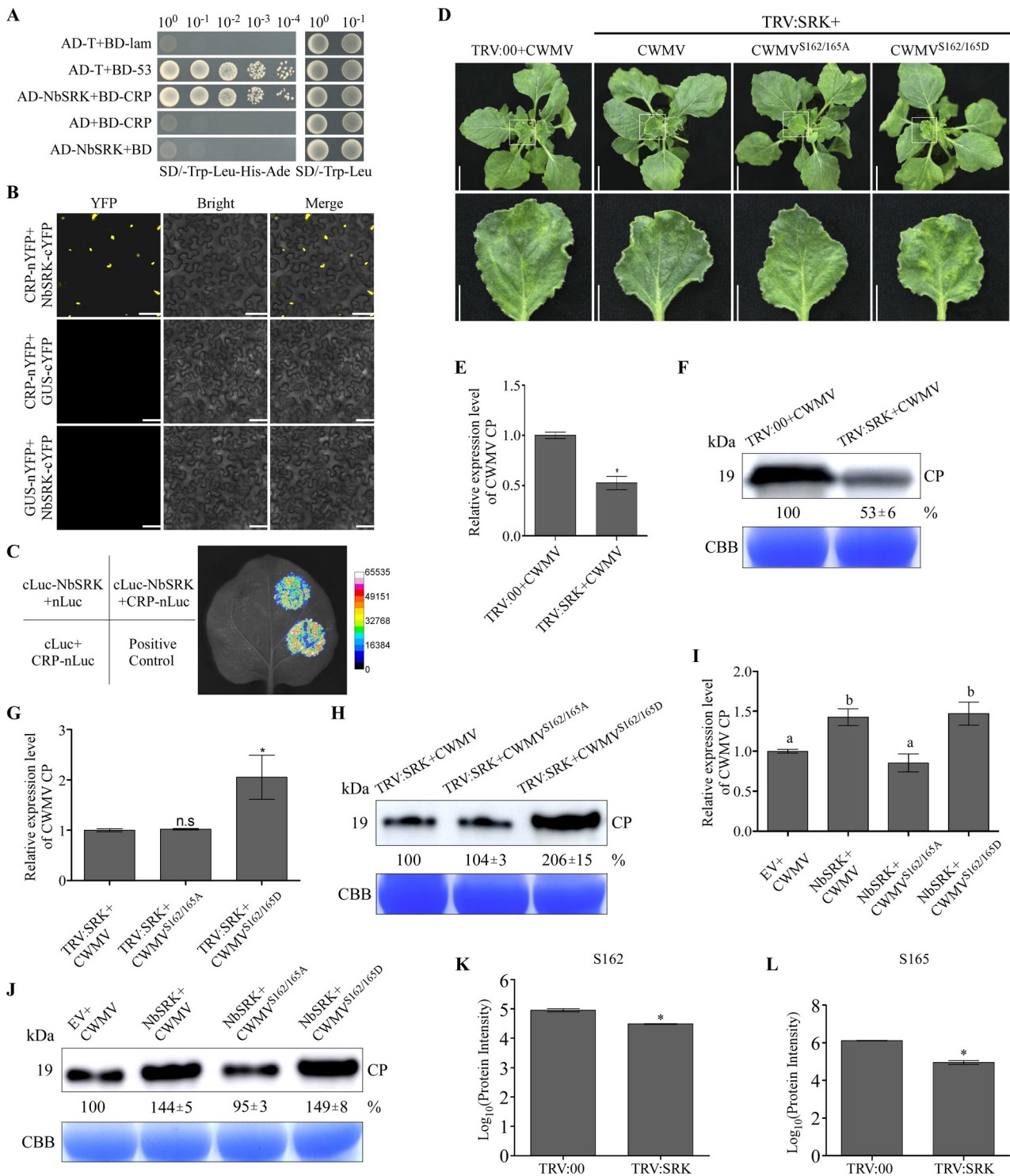

**Fig 4. SRK is required for CWMV infection in plant. A**, **B** and **C.** YTH, BiFC and LCI assay were used to confirm the interaction between NbSRK and CRP. **D.** Systemic mosaic symptoms on the TRV:00+CWMV-, TRV:SRK+CWMV-, TRV:SRK+CWMV$^{S162/165A}$-, or TRV:SRK+CWMV$^{S162/165D}$-inoculated *N. benthamiana* plants. Photographs were taken at 21 days post CWMV inoculation. Scale bar = 5 cm (upper panel), Scale bar = 2 cm (lower panel). **E.** Quantitative RT-PCR analysis of relative expression level of CWMV CP in the TRV:00+CWMV- or TRV:SRK+CWMV-inoculated *N. benthamiana* plants. The data presented are the means ± SD, determined using the Student's *t*-test. Each treatment had three biological replicates. *, $P < 0.05$. **F.** The accumulation of CWMV CP in the assayed *N. benthamiana* plants was determined through western blot analysis using a CWMV CP specific antibody at 21 dpi. The CBB-stained gel is used to show sample loadings. **G.** Relative expression levels of CWMV CP in the TRV:SRK +CWMV-, TRV:SRK+CWMV$^{S162/165A}$-, or TRV:SRK+CWMV$^{S162/165D}$-inoculated *N. benthamiana* plants were determined through qRT-PCR. The

data presented are the means ± SD, determined using the Student's *t*-test. Each treatment had three biological replicates. *, *P* <0.05; n.s, no significant difference. **H.** The accumulation of CWMV CP in the assayed *N. benthamiana* plants was determined through western blot analysis at 21 dpi. The CBB-stained gel is used to show sample loadings. **I.** Relative expression levels of CWMV CP in the EV+CWMV-, NbSRK+CWMV-, NbSRK +CWMV$^{S162/165A}$- or NbSRK+CWMV$^{S162/165D}$-inoculated *N. benthamiana* plants were determined through qRT-PCR. The data presented are the means ± SD, determined through the Tukey's test (*P* < 0.05). Each treatment had three biological replicates. **J.** Accumulation of CWMV CP in the assayed *N. benthamiana* plants was determined through western blot analysis using a CWMV CP specific antibody at 21 dpi. The CBB was used as loading controls. **K** and **L.** Quantifications of phosphorylation at S162 and S165. CRP-GFP fusion was expressed in the leaves of the TRV:00- or TRV: SRK-inoculated *N. benthamiana* plants followed by immunoprecipitation using an anti-GFP antibody. The phosphorylated peptides were identified through LC-MS/MS using the PRM method. Peptide phosphorylation ratios (e.g., phosphorylated/non-phosphorylated) were then determined. The data presented are the means ± SD, determined using the Student's *t*-test. Each treatment had three biological replicates. *, *P* <0.05.

NbSRK+CWMV$^{S162/165D}$-inoculated plants compared to the control plants, while the accumulation level of CWMV CP in the NbSRK+CWMV$^{S162/165A}$-inoculated plants was similar to that in the EV+CWMV-inoculated plants (Fig 4I). Further analysis of CWMV CP accumulation in these samples through western blot assays yielded a similar result (Fig 4J), indicating that overexpression of SRK can promote CWMV infection through phosphorylation of CRP at S162 and S165.

To elucidate the importance of NbSRK in CRP phosphorylation, CRP-GFP was transiently over-expressed in the TRV:00- or TRV:NbSRK-inoculated plants. At 3 dpi, CRP-GFP fusion was isolated through immunoprecipitation and then subjected to LC-MS/MS analysis. Phosphorylation of CRP at S162 and S165 was monitored through a parallel reaction monitoring (PRM) system. The result showed that the phosphorylation of CRP at S162 and S165 in the TRV:NbSRK-inoculated plants was significantly reduced compared to the TRV:00-inoculated control plants (Fig 4K and 4L), indicating that silencing of *NbSRK* expression affected the phosphorylation of CRP.

## CRP$^{S162/165D}$ can interact with TaUBA2C

In order to clarify the molecular mechanism of how S162 and S165 of CRP phosphorylation promotes CWMV infection, we screened a wheat cDNA library through YTH using CRP$^{S162/165D}$ as the bait. The results from three independent screenings showed that wheat UBP1-associated protein 2C (TraesCS3A02G220400.1, named as TaUBA2C thereafter) interacted with CRP$^{S162/165D}$. Blast searching *T. aestivum* database (http://plants.ensembl.org/) identified three *TaUBA2C* sequences (TraesCS3A02G220400.1, TraesCS3B02G250700.1, and TraesCS3D02G232300.1) (S5A Fig). Because these three sequences shear 98.06% sequence identity, we decided to use TraesCS3A02G220400.1 in further studies. Through YTH, we have found that CRP$^{S162D}$, CRP$^{S165D}$ and CRP$^{S162/165D}$, but not CRP$^{S162/165A}$, interacted with TaUBA2C, indicating that phosphorylation of CRP at S162 and/or S165 is crucial for the interaction (Fig 5A). This finding is supported by the results from the microscale thermophoresis (MST) and pull-down assays (Fig 5B and 5C). While MST assays show an interaction of TaUBAC with CRP$^{S162/165A}$, the different results obtained in different interaction experiments may be due to different experimental conditions. Because TaUBA2C contains two RRMs (Fig 5D), we analyzed the roles of these two RRMs via YTH using CRP$^{S162/165D}$ as the bait. The result showed that the second RRM (TaUBA2C$^{D2}$) was responsible for the interaction (Fig 5E).

## UBA2C can regulate CWMV infection in wheat

qRT-PCR analysis using tissues from the CWMV-infected or non-infected wheat plants showed that the expression of *TaUBA2C* was significantly up-regulated by CWMV infection (Fig 6A). To investigate the role of TaUBA2C on CWMV infection in wheat plants, we first silenced *TaUBA2C* expression using a BSMV-based VIGS vector. qRT-PCR at 10 dpi showed

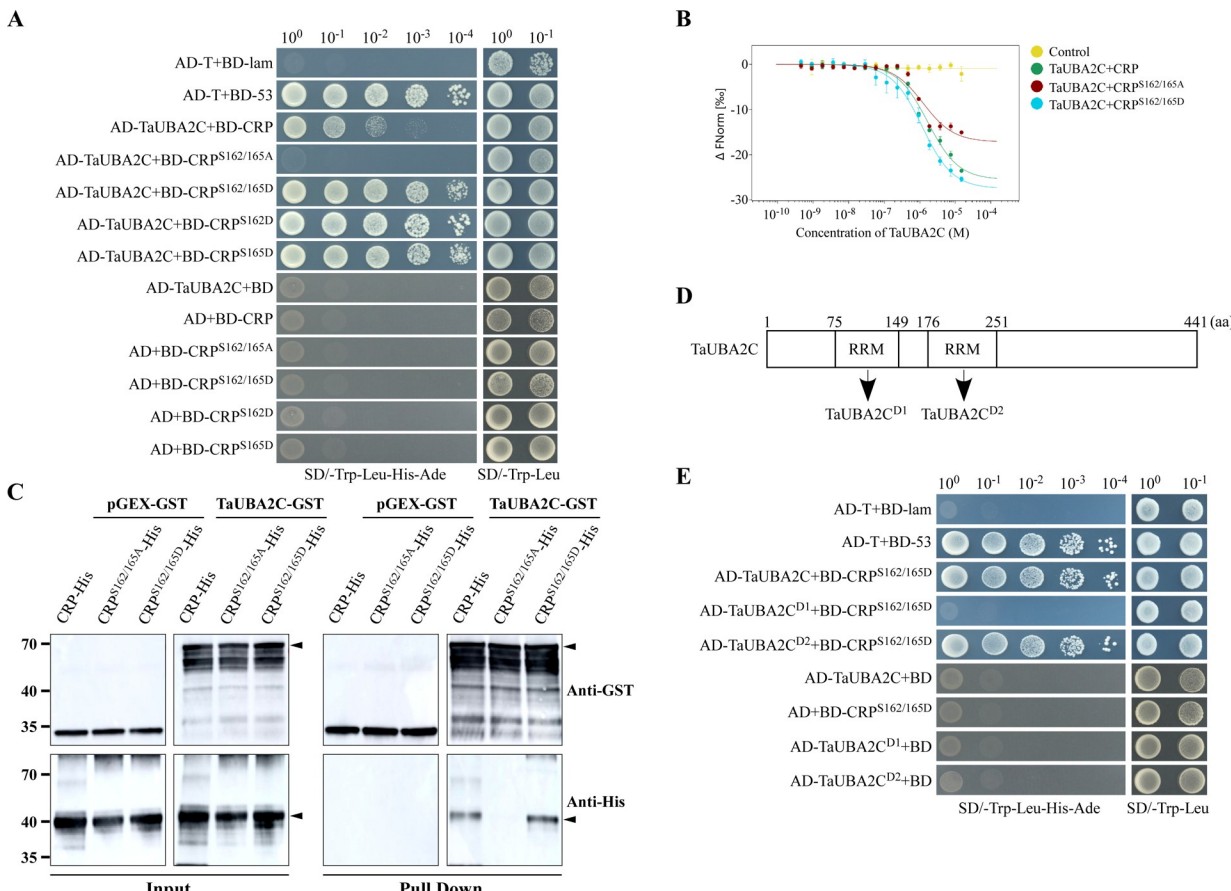

**Fig 5. CWMV CRP$^{S162/165D}$ can interact with TaUBA2C. A.** YTH assay was used to detect the interaction between TaUBA2C and CRP, CRP$^{S162/165A}$, CRP$^{S162/165D}$, CRP$^{S162D}$ or CRP$^{S165D}$. AD-TaUBA2C and BD-CRP or one of its derivatives were co-expressed in yeast cells. The transformed yeast cells were grown on the SD/-Leu/-Trp medium and then on the SD/-Trp/-Leu/-His/-Ade medium. Yeast cells co-expressing AD-T+BD-Lam or AD-T+BD-53 were used as controls. **B.** Microscale thermophoresis assay was used to detect the binding affinity of TaUBA2C to CRP, CRP$^{S162/165A}$ or CRP$^{S162/165D}$. Three independent experiments were conducted in this study and yielded similar results. Bars represent standard errors. **C.** Pull-down assay was used to confirm the *in vitro* interaction between TaUBA2C-GST and CRP-His, CRP$^{S162/165A}$-His or CRP$^{S162/165D}$-His. In this study, GST was used as a control. Black arrowheads indicate the positions of TaUBA2C-GST, CRP-His, CRP$^{S162/165A}$-His, and CRP$^{S162/165D}$-His, respectively. **D.** A schematic shows the arrangement of domains in TaUBA2C. The numbers above the schematic indicate the amino acid positions of different domains. RRM, RNA-recognition motif; aa, amino acids. **E.** YTH assay was used to detect the interaction between CRP and TaUBA2C, TaUBA2C$^{D1}$ or TaUBA2C$^{D2}$.

that the expression of *TaUBA2C* in the BSMV:TaUBA2C-inoculated plants was reduced by about 50% (S6A Fig). We then inoculated the BSMV:TaUBA2C- or BSMV:00-inoculated plants with CWMV (BSMV:TaUBA2C+CWMV or BSMV:00+CWMV). After 21 days post CWMV inoculation, stronger mosaic symptoms were observed on the BSMV:TaUBA2C +CWMV-inoculated plants than the BSMV:00+CWMV-inoculated plants (Fig 6B). The result of qRT-PCR showed that the accumulation level of CWMV CP in the BSMV:TaUBA2C +CWMV-inoculated plants was much higher than that in the BSMV:00+CWMV-inoculated plants (Fig 6C). Western blot result showed that the accumulation level of CWMV CP in the BSMV:TaUBA2C+CWMV-inoculated plants was much higher than that in BSMV:00 +CWMV-inoculated plants (Fig 6D), indicating that TaUBA2C has a key role in CWMV infection in wheat.

To further validate this finding, we generated several transgenic wheat lines overexpressing TaUBA2C-His. Analysis these lines through western blot assay showed that line 3 and line 5

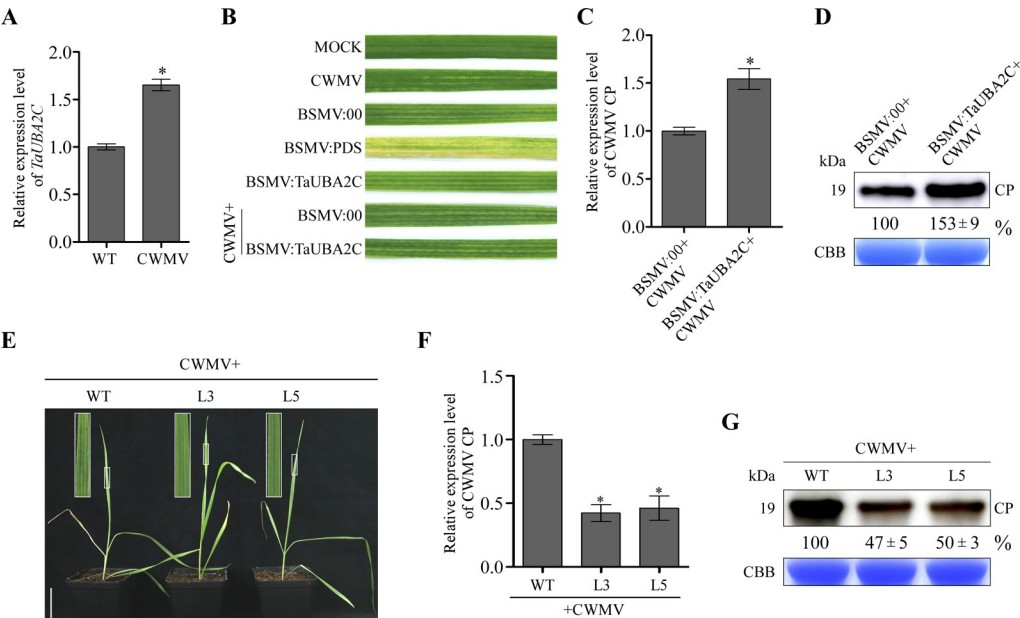

**Fig 6. TaUBA2C can facilitate CWMV infection. A.** Analysis of *TaUBA2C* expression in the mock- or the CWMV-inoculated wheat plants through qRT-PCR. The data presented are the means ± SD, determined using the Student's *t*-test. Each treatment had three biological replicates. *, *P* <0.05. **B.** Mosaic symptoms in the wheat leaves infected with CWMV, BSMV:00, BSMV:PDS, BSMV:TaUBA2C, BSMV:00+CWMV or BSMV:TaUBA2C+CWMV. Mock indicates that the leaf was from a plant inoculated with the FES buffer. Photographs were taken at 21 dpi. **C.** Quantitative RT-PCR analysis of CWMV CP accumulation in the BSMV:00+CWMV- or BSMV:TaUBA2C+CWMV-inoculated wheat plants at 21 dpi. The data presented are the means ± SD, determined using the Student's *t*-test. Each treatment had three biological replicates. *, *P* <0.05. **D.** Detection of CWMV CP in the assayed wheat plants through western blot analysis using a CWMV CP specific antibody. The data presented are the means ± SD. Each treatment had three biological replicates. The CBB-stained gel is used to show sample loadings. **E.** Disease symptoms on the CWMV-inoculated wild type (WT) or the two TaUBA2C overexpression transgenic lines (L3 and L5). Photographs were taken at 21 days post CWMV inoculated. Scale bar = 5 cm. **F.** Detection of CWMV CP accumulation in the CWMV-inoculated WT, L3, and L5 plants, respectively, through qRT-PCR at 21 dpi. The data presented are the means ± SD, determined using the Student's *t*-test. Each treatment had three biological replicates. *, *P* <0.05. **G.** Detection of CWMV CP accumulation in the assayed plants through western blot analysis using a CWMV CP specific antibody. The data presented are the means ± SD. Each treatment had three biological replicates. CBB-stained gel is used to show sample loadings.

(L3 and L5) plants produced more TaUBA2C-His than other line plants (S6B Fig). We then inoculated the seedlings of L3 and L5 with *in vitro* transcribed CWMV RNAs. After 21 days, the CWMV-inoculated L3 and L5 plants showed milder mosaic symptoms than the CWMV-inoculated wild type (WT) wheat plants (Fig 6E). The results of qRT-PCR and western blot analysis revealed that the accumulation level of CWMV CP were significantly decreased in the two CWMV-inoculated transgenic lines compared to that in the CWMV-inoculated WT control plants (Fig 6F and 6G), indicating that TaUBA2C is regulator of CWMV infection.

## Phosphorylation of CRP suppresses cell death induced by TaUBA2C

Transient expression of *AtUBA2*s in *Arabidopsis* leaves caused cell death [38]. To investigate whether TaUBA2C is also an inducer of cell death, we transiently expressed TaUBA2C-Flag in *N. benthamiana* leaves through agroinfiltration (S7A Fig). The leaves expressing PVX-bax were used as positive controls. Trypan blue staining result showed that at 5 dpi, the TaU-BA2C-Flag or PVX-bax expressing *N. benthamiana* leaves developed cell death symptom (Fig 7A). To investigate the role of CRP and TaUBA2C interaction on cell death induction, we transiently co-expressed CRP-HA, CRP$^{S162/165A}$-HA or CRP$^{S162/165D}$-HA with TaUBA2C-Flag

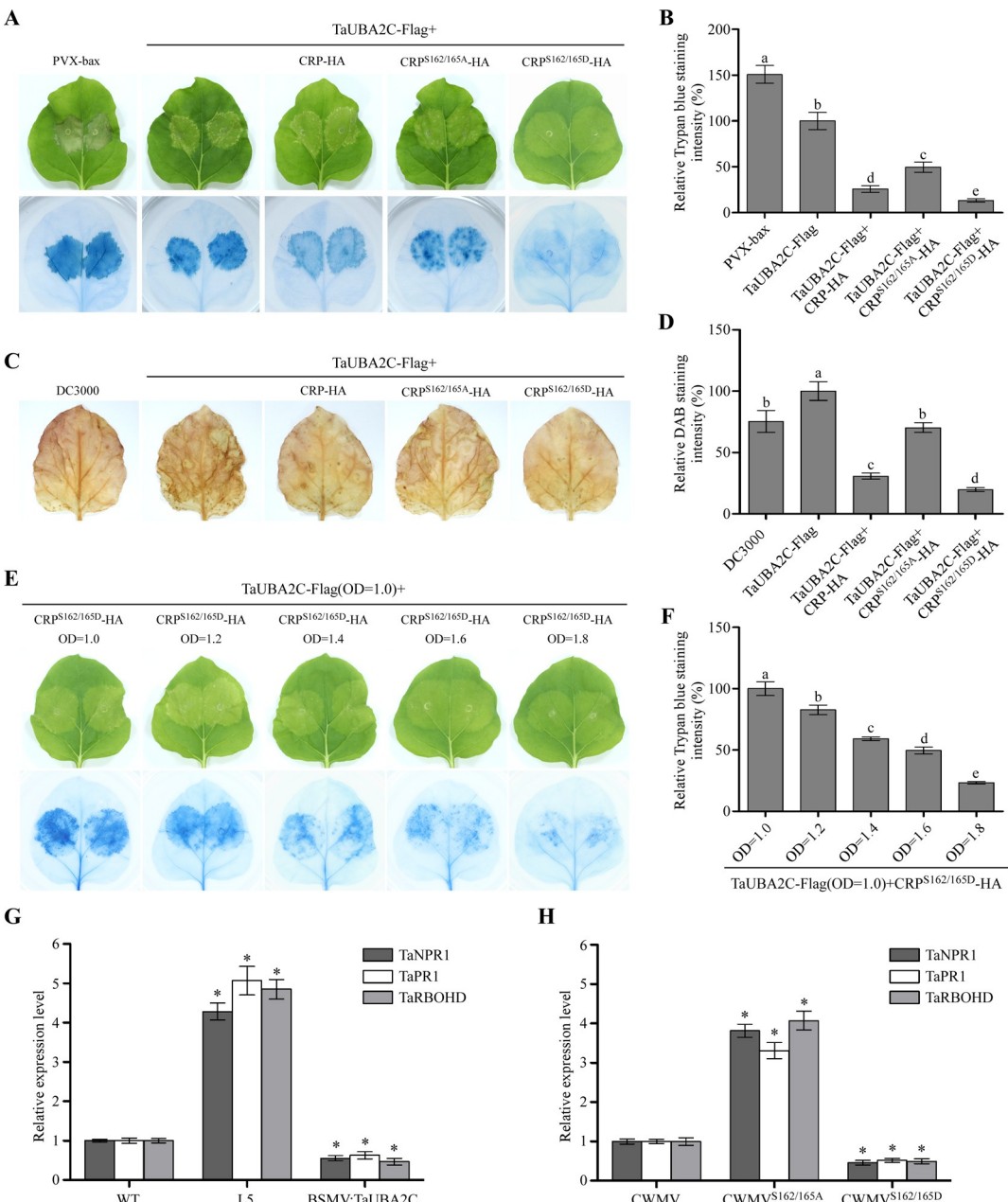

**Fig 7. Phosphorylation of CRP at S162 and S165 suppresses cell death induced by TaUBA2C. A.** Cell death in *N. benthamiana* leaf tissues expressing TaUBA2C, TaUBA2C+CRP, TaUBA2C+CRP$^{S162/165A}$ or TaUBA2C+CRP$^{S162/165D}$. The infiltrated leaves were harvested and photographed at 5 days post agroinfiltration (upper panel), and then stained with a Trypan blue solution (lower panel). An *N. benthamiana* leaf inoculated with PVX-bax was used as a positive control. **B.** Measurement of the relative Trypan blue staining intensity shown in Fig 7A. Each treatment had three biological replicates, the data presented are the means ± SD. Different letters show statistically significant differences (P < 0.05, Tukey's test). **C.** Assayed *N. benthamiana* leaves were stained with a DAB solution at 5 dpi. The *N. benthamiana* leaf inoculated with the wild type Agrobacterium DC3000 was used as a control. **D.** Measurement of the relative DAB staining intensity shown in Fig 7C. Each treatment had three biological replicates, the data presented are the means ± SD. Different letters show statistically significant differences (P < 0.05, Tukey's test). **E.** *N. benthamiana* leaves co-inoculated with TaUBA2C and different concentrations of CRP$^{S162/165D}$ were harvested and photographed at 5 dpi (upper panel), and then stained with a Trypan blue solution (lower panel). **F.** Measurement of the relative Trypan blue staining intensity shown in Fig 7E. Each treatment had three biological replicates, the data presented are the means ± SD. Different letters show statistically significant differences (P < 0.05, Tukey's test). **G.** Quantitative RT-PCR analyses of *TaNPR1*, *TaPR1* and *TaRBOHD* expressions in the wild type (WT), L5 transgenic and BSMV:TaUBA2C-inoculated plants. The data presented are the means ± SD, determined using the Student's *t*-test. Each treatment had three biological replicates. *, P <0.05. **H.** Quantitative RT-PCR analyses of *TaNPR1*,

*TaPR1* and *TaRBOHD* expressions in the CWMV-inoculated, CWMV$^{S162/165A}$-inoculated or CWMV$^{S162/165D}$-inoculated plants. The data presented are the means ± SD, determined using the Student's *t*-test. Each treatment had three biological replicates. $^*$, $P < 0.05$.

in *N. benthamiana* leaves through agroinfiltration (S7A Fig). Trypan blue staining result showed that the *N. benthamiana* leaves co-expressing TaUBA2C-Flag and CRP-HA (TaUBA2C-Flag+CRP-HA) or TaUBA2C-Flag and CRP$^{S162/165D}$-HA (TaUBA2C-Flag+CRP$^{S162/165D}$-HA) showed less cell death than that in the leaves expressing TaUBA2C-Flag alone (Fig 7A and 7B). In this study, the leaves co-expressing TaUBA2C-Flag and CRP$^{S162/165A}$-HA (TaUBA2C-Flag+CRP$^{S162/165A}$-HA) showed moderate cell death (Fig 7A and 7B), indicating that phosphorylation of CRP can suppress the TaUBA2C-induced cell death in plant.

Because cell death is related to $H_2O_2$ production, we analyzed the accumulation level of $H_2O_2$ in various *N. benthamiana* leaves through DAB staining. The result showed that less $H_2O_2$ had accumulated in the leaves co-expressing TaUBA2C-Flag+CRP-HA or TaUBA2C-Flag+CRP$^{S162/165D}$-HA compared to the leaves expressing TaUBA2C-Flag alone (Fig 7C and 7D). The leaves co-expressing TaUBA2C-Flag+CRP$^{S162/165A}$-HA accumulated moderate level of $H_2O_2$ (Fig 7C and 7D). In the next experiment, we co-expressed TaUBA2C-Flag and different concentrations of CRP$^{S162/165D}$-HA in *N. benthamiana* leaves (S7C Fig). The result of Trypan blue staining showed that the effect of CRP$^{S162/165D}$-HA on TaUBA2C-induced cell death was indeed dose-dependent (Fig 7E and 7F). In addition, we analyzed the effect of TaUBA2C on the expressions of defense-related genes and $H_2O_2$ biogenesis-related genes. Total RNA was isolated from the WT, TaUBA2C transgenic and BSMV:TaUBA2C-inoculated plant tissues, respectively, and analyzed for the expressions of *TaNPR1*, *TaPR1* and *TaRBOHD* through qRT-PCR. The results showed that the expressions of these three genes were significantly up-regulated in the TaUBA2C transgenic plants, but significantly down-regulated in the BSMV: TaUBA2C-inoculated plants (Fig 7G), suggesting that TaUBA2C may regulate the expressions of *TaNPR1*, *TaPR1* and *TaRBOHD*. Moreover, we studied the expressions of these three target genes in CWMV-inoculated, CWMV$^{S162/165A}$-inoculated and CWMV$^{S162/165D}$-inoculated plants. The results showed that the expressions of these three genes were significantly up-regulated in the CWMV$^{S162/165A}$-inoculated plants, but significantly down-regulated in the CWMV$^{S162/165D}$-inoculated plants (Fig 7H), indicating that the pathogenicity of CWMV is closely related to the expression levels of three target genes. Based on the above findings we conclude that phosphorylation of CRP at S162 and S165 can attenuate the TaUBA2C-induced cell death.

## Phosphorylated CRP disrupts the RNA- and DNA-binding activity of TaUBA2C

To analyze the subcellular localization of TaUBA2C, we first expressed TaUBA2C-CFP in the leaves of the H2B-RFP transgenic *N. benthamiana* plants through agroinfiltration. Under the confocal microscope, TaUBA2C-CFP fusion appeared as speckles in the nucleus at 2 dpi (Fig 8A). Consistent with this result, the predictions from the online servers (http://nls-mapper.iab.keio.ac.jp/cgi-bin/NLS_Mapper_form.cgi) suggested that there is a nuclear localization signal sequence (MDPFSKKRKPDE) in the TaUBA2C sequence. CWMV CRP is known to accumulate in the cytoplasm [32]. In this study, we also found that CRP$^{S162/165A}$-GFP and CRP$^{S162/165D}$-GFP accumulated in the cytoplasm (S8A Fig). To investigate where CRP interacts with TaUBA2C in cell, we conducted BiFC assay in leaves of H2B-RFP transgenic *N. benthamiana* plants. The result showed that the interaction between CRP-nYFP and TaUBA2C-cYFP occurred in nucleus with less defined speckles (Fig 8B). Similar result was also obtained for the

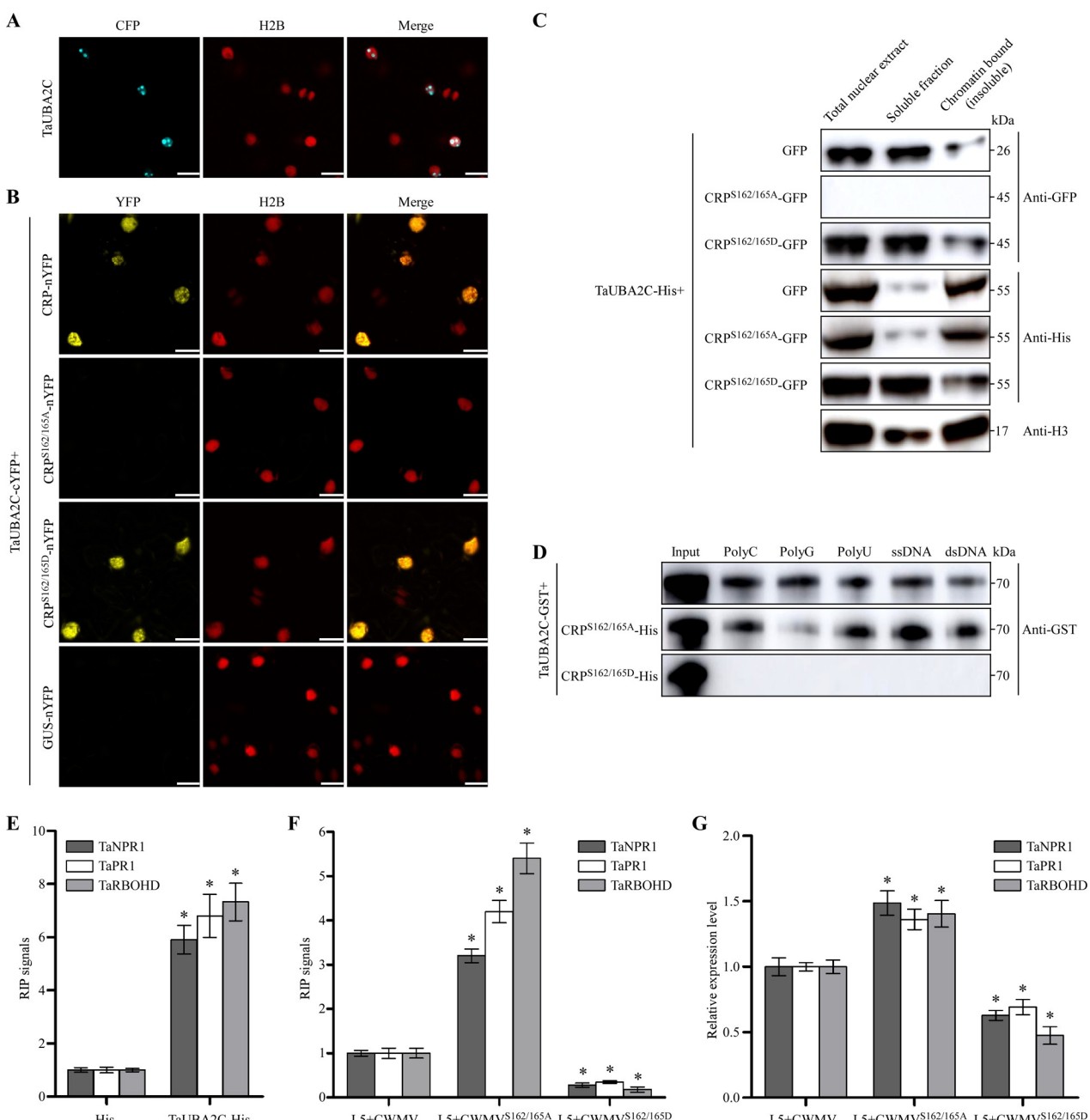

**Fig 8. Phosphorylated CRP can interfere TaUBA2C RNA and DNA binding activities. A.** Subcellular localization of TaUBA2C-CFP in H2B-RFP transgenic *N. benthamiana* plants epidermal cells. Confocal images were taken at 60 h post infiltration (hpi). Scale bar = 20 μm. **B.** Analysis of subcellular localization of the co-expressed TaUBA2C-cYFP and CRP-nYFP, TaUBA2C-cYFP and CRP$^{S162/165A}$-nYFP, TaUBA2C-cYFP and CRP$^{S162/165D}$-nYFP, and TaUBA2C-cYFP and GUS-cYFP, respectively, in H2B-RFP transgenic *N. benthamiana* leaves through BiFC assay. Confocal images were taken at 60 hpi. Scale bar = 20 μm. **C.** Western blot analyses of subcellular localizations of the co-expressed TaUBA2C-His and GFP, TaUBA2C-His and CRP$^{S162/165A}$-GFP, or TaUBA2C-His and CRP$^{S162/165D}$-GFP using an anti-GFP or an anti-His antibody. Histone H3 protein was used as a chromatin-bound protein marker. **D.** RNA and DNA binding activity assays. TaUBA2C-GST, CRP$^{S162/165A}$-His and CRP$^{S162/165D}$-His were individually expressed and purified. The purified TaUBA2C-GST was first mixed with CRP$^{S162/165A}$-His or CRP$^{S162/165D}$-His, and then mixed with polycytidylic [poly(rC)], polyguanylic [poly(rG)], polyuridylic [poly(rU)], single-stranded DNA or double-stranded DNA. After incubation, these samples were analyzed through western blot assays using an anti-GST antibody. **E.** RIP-RT-qPCR showing the enrichment of *TaNPR1*, *TaPR1* and *TaRBOHD* pre-mRNA on protoplasts expressing TaUBA2C-His. The protoplasts expressing His were used as negative controls. RIP signals were presented as the percentage of immunoprecipitated *TaNPR1*, *TaPR1* and *TaRBOHD* pre-mRNA relative to input *TaNPR1*, *TaPR1* and *TaRBOHD* pre-mRNA. The data presented are the means ± SD, determined using the Student's *t*-test. Each treatment had three biological replicates. *, *P* <0.05. **F.** RIP-RT-qPCR showing the enrichment of *TaNPR1*, *TaPR1* and *TaRBOHD* pre-mRNA on TaUBA2C-His transgenic wheat seedlings infected with CWMV, CWMV$^{S162/165A}$ or CWMV$^{S162/165D}$, respectively. The data presented are the means ± SD, determined using the Student's *t*-test. Each

treatment had three biological replicates. *, $P$ <0.05. **G.** Quantitative RT-PCR analyses of *TaNPR1*, *TaPR1* and *TaRBOHD* expressions in the CWMV-inoculated, CWMV$^{S162/165A}$-inoculated or CWMV$^{S162/165D}$-inoculated TaUBA2C-His transgenic wheat seedlings. The data presented are the means ± SD, determined using the Student's *t*-test. Each treatment had three biological replicates. *, $P$ <0.05.

interaction between CRP$^{S162/165D}$-nYFP and TaUBA2C-cYFP. In contrast, no YFP fluorescence was observed in the leaf cells co-expressing TaUBA2C-cYFP and CRP$^{S162/165A}$-nYFP or TaUBA2C-cYFP and GUS-nYFP (control), indicating that the interaction between CRP and TaUBA2C or CRP$^{S162/165D}$-nYFP and TaUBA2C inhibited the formation of TaUBA2C speckles in nucleus. Based on the above results, we speculated that the entry of CRP into the nucleus depended on the interaction of CRP and TaUBA2C in cell. Therefore, we performed BiFC analysis with TaUBA2C$^{ΔNLS}$-cYFP without the nuclear localized signal peptide and CRP-nYFP or CRP$^{S162/165D}$-nYFP in leaves of H2B-RFP transgenic *N. benthamiana* plants. We could observe that the interaction between TaUBA2C$^{ΔNLS}$-cYFP and CRP-nYFP or CRP$^{S162/165D}$-nYFP occurred in cytoplasm but not in the nucleus (S9 Fig). The results suggest that CRP might be brought into the nucleus by interaction with TaUBA2C. We then co-expressed GFP, CRP$^{S162/165A}$-GFP or CRP$^{S162/165D}$-GFP with TaUBA2C-His in *N. benthamiana* leaves through agroinfiltration followed by protein extraction at 3 dpi. Western blot results showed that by 3 dpi and in the leaves co-expressing TaUBA2C-His and GFP, TaUBA2C-His was mainly present in the nuclear insoluble fraction. Similar result was found for the leaves co-expressing TaUBA2C-His and CRP$^{S162/165A}$-GFP. In the leaves co-expressing TaUBA2C-His and CRP$^{S162/165D}$-GFP, however, TaUBA2C-His was found mostly in the nuclear soluble fraction (Fig 8C). Because histone H3 is associated with chromatin and is present in the nuclear insoluble fraction [39], we speculate that TaUBA2C is likely a chromatin-bound protein, and the interaction with CRP$^{S162/165D}$ can change its chromatin-bound status.

RRM-containing proteins are known to have RNA- or DNA-binding activities [40–42]. To determine whether TaUBA2C can also bind to RNA or DNA, we performed RNA- and DNA-binding assays. The results showed that TaUBA2C was indeed capable of binding to RNA polymers [e.g., poly(rC), poly(rG) and poly(rU)], single- and double-stranded DNA (Fig 8D). After TaUBA2C-GST was mixed with CRP$^{S162/165D}$-His, and incubated with the immobilized glutathione resins, however, the RNA- and DNA-binding activity of TaUBA2C was lost (Fig 8D).

We then performed RNA-immunoprecipitation (RIP) assays to investigate whether TaUBA2C associates with *TaNPR1*, *TaPR1* and *TaRBOHD* pre-mRNA. The plasmid expressing His (control) or TaUBA2C-His fusion were introduced into wheat protoplasts via PEG 4000-based method. After overnight incubation at room temperature, total protein was extracted from the inoculated protoplasts. Then, we used the His antibody to immunoprecipitate TaUBA2C-His from total protein extracts. The resulting immunoprecipitates of TaUBA2C-His were reverse-transcribed into cDNA and measured by RT-qPCR using specific primers for *TaNPR1*, *TaPR1* and *TaRBOHD*, respectively. We present the RIP signal as the percentage of immunoprecipitated *TaNPR1*, *TaPR1* and *TaRBOHD* pre-mRNA relative to input *TaNPR1*, *TaPR1* and *TaRBOHD* pre-mRNA. As shown in Fig 8E, the pre-mRNA of *TaNPR1*, *TaPR1* and *TaRBOHD* were abundantly detected in immunoprecipitates from the protoplasts expressing TaUBA2C-His, indicating that TaUBA2C indeed associates with *TaNPR1*, *TaPR1* and *TaRBOHD* pre-mRNA. Next, we also investigate whether CRP$^{S162/165D}$ can affect TaUBA2C associates with *TaNPR1*, *TaPR1* and *TaRBOHD* pre-mRNA. TaUBA2C-His transgenic line was infected with CWMV, CWMV$^{S162/165A}$ or CWMV$^{S162/165D}$, respectively. Interestingly, RIP signals were substantially decreased in the CWMV$^{S162/165D}$-inoculated plants, while RIP

signals were increased in the CWMV$^{S162/165A}$-inoculated plants (Fig 8F). Collectively, our results suggesting that the association of TaUBA2C with the *TaNPR1*, *TaPR1* and *TaRBOHD* pre-mRNA could be reduced by the CRP$^{S162/165D}$ bind to TaUBA2C. Then we also investigate the expressions of these three target genes in TaUBA2C-His transgenic lines infected with CWMV, CWMV$^{S162/165A}$ or CWMV$^{S162/165D}$, respectively. As expected, the expressions of these three genes were significantly up-regulated in the CWMV$^{S162/165A}$-inoculated TaUBA2-C-His transgenic plants, but significantly down-regulated in the CWMV$^{S162/165D}$-inoculated TaUBA2C-His transgenic plants (Fig 8G). These results showed that phosphorylated CRP disrupts the RNA- and DNA-binding activity of TaUBA2C which in turn reduced the association of TaUBA2C with the *TaNPR1*, *TaPR1* and *TaRBOHD* pre-mRNA, thereby down-regulating the expressions of these three target genes.

## Discussion

Protein phosphorylation is a reversible post-translational process that modifies proteins to modulate gene expression, signal transduction, protein subcellular localization, and protein interactions [43]. Numerous studies have indicated that many viral proteins can be phosphorylated by host protein kinases and the phosphorylated viral proteins can alter virus pathogenicity in infected plants [4,5,12]. In this study, we have identified two phosphorylation sites (e.g., S162 and S165) in CWMV CRP (Fig 1). Through reverse genetics, we have found that CWMV$^{S162/165D}$, a mutant CWMV with a mimic phosphorylation CRP, can cause stronger virus infection and disease symptoms than the WT CWMV or CWMV$^{S162/165A}$, a mutant CWMV with a non-phosphorylatable CRP (Fig 2). Therefore, we consider that phosphorylation of CRP at S162 and S165 is important for CWMV infection in plant. Moreover, we have found that SAPK7 can bind to CRP and phosphorylate it at S162 and S165 to promote CWMV infection in plant. Interestingly, SAPK7 is a member of the plant SnRK2 protein kinase family which are mostly involved in plant responses to abiotic environmental stresses [44]. Gemini-viruses interact with multiple plant protein kinases, such as SnRK1, to modulate host signaling networks prompting viral infection [11,45,46]. Our findings suggest that SnRK2s can also be recruited by plant pathogens for enhancing their infection in the host. Furthermore, S156 and T159 were identified as secondary phosphorylation sites of CRP, whose phosphorylation depends on the phosphorylation of S162 and/or S165. However, our results indicated that phosphorylation of CRP at S156 and T159 does not affect the pathogenicity of CWMV (S3 Fig). Further research was needed to study the function of these two sites. Collectively, S162 and S165 of CRP are more important for the pathogenicity of CWMV. As expected, silencing *NbSRK* expression in *N. benthamiana* plant inhibited the phosphorylation of CRP at S162 and S165, leading to a reduced CWMV infection (Figs 3 and 4). CWMV CRP is an RNA silencing suppressor [32]. Previous studies have suggested that phosphorylation of VSRs can affect their RNA silencing suppression activities. For example, phosphorylation of CMV VSR, 2b, altered its subcellular localization pattern and reduced its VSR activity [47]. Phosphorylation of BSMV γb at S96 also affected its VSR activity [4]. Unlike CMV 2b and BSMV γb, we have found that phosphorylation of CWMV CRP at S162 and S165 had no clear effect on its VSR activity (S2 Fig). The difference between our result and the results published previously is likely because the S162 and S165 are located in the C-terminus of CRP, while the N-terminus and the central region of CRP are the key domains for its VSR activity [32]. Therefore, we consider that the mechanism underlying CRP phosphorylation-mediated CWMV infection is different from that controlling its VSR activity.

Cell death plays important roles in plant innate immunity to pathogen invasions [48]. To date, several RBPs have been found to induce cell death in plants. For instance, through

transient expression in tobacco leaves, potato UBP1-associated protein 2c (StUBA2c) has been found to induce a HR-like cell death [49]. CaRBP1, a RBP found in pepper, was also found to induce hypersensitive cell death and $H_2O_2$ production in peper leaves [50]. TaUBA2C is a member of RBP in wheat and has two RRM domains. Our result has shown that TaUBA2C can also induce cell death and $H_2O_2$ production in plant. Furthermore, our results also demonstrated that silencing of TaUBA2C expression in wheat significantly enhances host susceptibility to CWMV infection. In contrast, overexpression of TaUBA2C in wheat enhances host resistance to CWMV infection. Thus, we reasoned that TaUBA2C may be crucial for resistance to viral infection through inducing the cell death and $H_2O_2$ production responses. During evolution, pathogens have evolved different strategies to circumvent host cell death response. For example, *Pseudomonas syringae* encodes a type III effector to suppress host plant cell death response [51]. *Phytophthora infestans* has been reported to suppress programmed cell death in plant through secreting an effector protein (SNE1) [52]. After phosphorylation by PKA, BSMV γb protein can suppress host cell death to ensure virus systemic infection [4]. In this study, we have found that both WT CRP and CRP$^{S162/165D}$ can bind to TaUBA2C to attenuate the TaUBA2C-mediated cell death and $H_2O_2$ production. It is noteworthy that the phosphorylatable CRP but not the non-phosphorylatable CRP is more effective in inhibiting cell death and $H_2O_2$ production. Furthermore, this inhibitory effect is found to be CRP$^{S162/165D}$ dose-dependent (Fig 7). These results suggest that the interaction of CRP and TaUBA2C is necessary but not sufficient for inhibiting cell death and $H_2O_2$ production. However, phosphorylation of S162 and S165 increased its binding activity to TaUBA2C, possibly via phosphorylation induce CRP conformational changes.

Activation and proper function of proteins are dependent on their proper subcellular localizations. To regulate PR gene expressions, *Arabidopsis* NPR1 needs to be in nucleus [53]. To positively regulate the SA-mediated immunity, AtRBP1-DR1 needs to be in the cytoplasm [24]. Similarly, the cytoplasmic localization of CaRBP1 is crucial for the induction of a cell death response [50]. Here, we found TaUBA2C can form speckles in nucleus and is present mostly in the nuclear insoluble fraction. Nuclear speckles are viewed as storage and assembly areas that procure splicing factors to active transcription sites [54]. RNA-binding proteins with RRMs are also known to involve in post-transcriptional processes. For instance, *Arabidopsis* AtGRP7 has been shown to bind to its pre-mRNA to modulate its own expression through alternative RNA splicing [55]. Based on these findings, we hypothesize that TaUBA2C bind to the target RNAs to regulate gene expressions through a post-transcriptional modification manner. Regulation of defense gene expression via post-transcriptional modification is crucial for innate immunity [20]. In this study, we have found that the expressions of *TaNPR1*, *TaPR1* and *TaRBOHD* were significantly down-regulated in the TaUBA2C-silenced wheat plants, while significantly up-regulated in the TaUBA2C overexpression transgenic plants (Fig 7). *NPR1* gene encodes an important positive regulator of salicylic acid responses to activate many *PR* genes expression, including *PR1*, which is thus a good marker gene for activation of this pathway [53]. *RBOHD* play an important role in $H_2O_2$ production in response to pathogens [56]. Increased $H_2O_2$ levels may be a primary cause of hypersensitive cell death [48]. In addition, we have found that the interaction between CRP$^{S162/165D}$ and TaUBA2C can inhibit the formation of TaUBA2C speckles in nuclei, and disrupt its ability to bind RNA molecules. Moreover, the CWMV$^{S162/165D}$ but not the CWMV$^{S162/165A}$ infection could inhibit the expression of *TaNPR1*, *TaPR1* and *TaRBOHD* in the TaUBA2C overexpression transgenic plants (Fig 8). Collectively, we consider that the phosphorylation of CRP may reduce the TaUBA2C bind ability to certain target RNAs in post-transcriptional processes resulting in inhibition of cell death and $H_2O_2$ production.

In summary, through this study, we have found that during CWMV infection in plant, its CRP is phosphorylated by SAPK7. The phosphorylated CRP interacts with TaUBA2C and disassociates it from chromatin to disrupt its RNA and DNA binding activities, resulting in an attenuated cell death and an enhanced CWMV infection (Fig 9). Further research is needed to determine other targets of TaUBA2C and the molecular mechanism regulating the TaUBA2C-mediated post-transcriptional modification during CWMV infection.

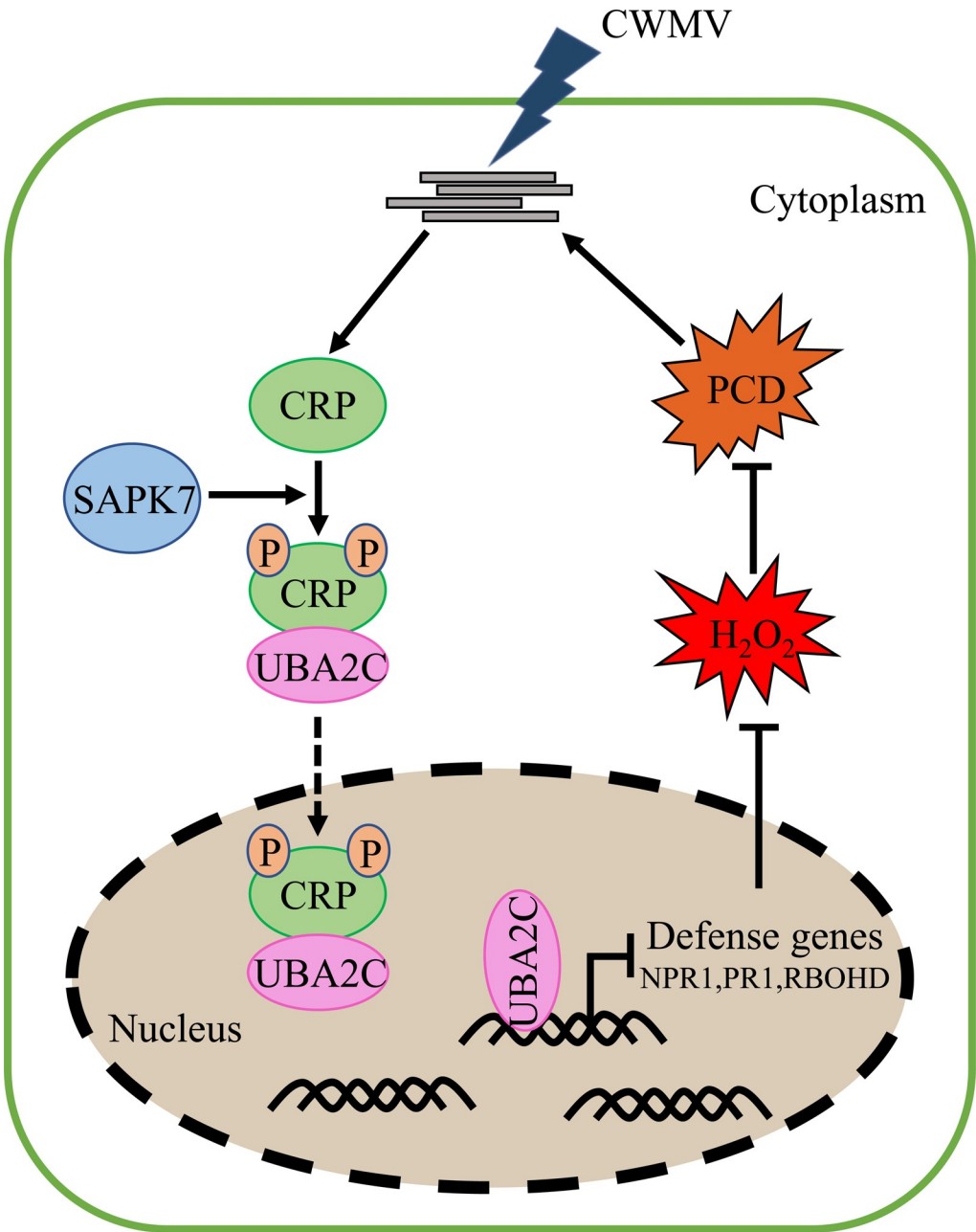

**Fig 9. Working model of SAPK7 phosphorylating CWMV CRP to promote viral infection.** SAPK7 phosphorylates CWMV CRP, phosphorylated CRP interactes with UBA2C, which changes TaUBA2C chromatin-bound status and attenuates its association with the *TaNPR1*, *TaPR1* and *TaRBOHD* pre-mRNA, leading the down-regulation expression of TaNPR1, TaPR1 and TaRBOHD. The decrease of these defense genes inhibit $H_2O_2$ production and cell death, which in turn promotes virus infection.

## Materials and methods

### Plant growth and virus inoculation

*N. benthamiana* and wheat cv. Yangmai 158 plants were grown in pots inside a greenhouse maintained at 25˚C, 65±5% relative humidity, and a 16-hour light/8-hour dark photoperiod. After inoculation with CWMV, CWMV$^{S162/165A}$ or CWMV$^{S162/165D}$, the plants were grown inside a climate room maintained at 15˚C until further analysis. For BSMV inoculation, wheat seedlings were inoculated at two-leaf stage and then grown inside a climate growth chamber maintained at 28–30˚C. The CWMV and PVX-bax infectious clones were obtained from a previously reported source [30]. Plasmid pCB-35S-R1 and pCB-35S-R2, containing full-length CWMV RNA1 or RNA2 sequence, were individually transformed into *Agrobacterium tumefaciens* strain GV3101. The Agrobacterium cultures were grown overnight, pelleted through centrifugation, and then incubated for 2 hours in an infiltration buffer (100 mM MES, pH 5.2, 10 mM MgCl$_2$, and 200 mM acetosyringone). Agrobacterium culture carrying pCB-35S-R1 (OD$_{600}$ = 0.6–0.8) was mixed with an equal volume of Agrobacterium culture carrying pCB-35S-R2 or one of its mutants. The mixed cultures were infiltrated individually into leaves of *N. benthamiana* plants using needleless syringes. Inoculation of wheat seedlings with *in vitro* transcribed CWMV RNAs was as reported previously [57]. Briefly, plasmids were individually linearized with *Spe*I restriction enzyme followed by *in vitro* transcription using the Ambion Message Machine kit (Invitrogen). *In vitro* transcribed CWMV RNA1 transcript (5 μg) was mixed with an equal amount of transcript representing CWMV RNA2 or its mutants. The mixed RNA transcripts were individually diluted in 1× FES buffer (0.06 M K2HPO4, 0.1 M glycine, 1% tetrasodium pyrophosphate, 1% celite, and 1% bentonite in nuclease-free water, pH 8.5) followed by rub-inoculation to the second true leaf of each wheat seedling (10 μL per leaf). All experiments were repeated at least 3 times, and each experiment included at least 6 plants. CWMV infection rate is 100%. Symptoms in *N. benthamiana* or wheat plants infected with the WT CWMV or its mutants were photographed at 21 dpi, according to previously reported [58]. Symptoms are classified based on the degree of yellowing and curling of the leaves, as shown in the S10 Fig, the symptoms gradually increase from I to IV.

### Plasmid construction

Seven mutant plasmid pCWMV$^{S162A}$, pCWMV$^{S165A}$, pCWMV$^{S162/165A}$, pCWMV$^{S162D}$, pCWMV$^{S165D}$, and pCWMV$^{S162/165D}$ were generated through overlapping PCR using plasmid pCB-35S-R2 as the template. CWMV *CRP* gene sequence was amplified and cloned into the pGWB5C, pET-32a, pGBKT7, pCV-nYFP, pCambia1300-nLUC, pGWB514 and pGWB408 vectors, respectively. The coding sequence of NbSRK, TaSAPK7 or TaUBA2C was cloned into the pGADT7, pGEX-4T-2, pCV-cYFP, pGWB511, and pCambia1300-cLUC vectors, respectively. The primers used in this study are listed in the S3 Table.

### Wheat protoplast isolation and transfection

The protoplasts were isolated from the 2-week-old wheat seedlings as previously described [57]. The plasmid GFP empty vector or CRP-GFP was transfected into wheat protoplasts respectively, using a PEG-mediated transformation method (Bio-Rad, Hercules, CA, USA). The transfected protoplasts were washed twice using W5 solution and incubated at 25˚C for 16–18 hours. Then, the protoplasts were harvested for protein extraction.

## RNA extraction and quantitative reverse transcription polymerase chain reaction (qRT-PCR)

Total RNA was extracted from *N. benthamiana* or wheat tissue samples using the HiPure plant RNA mini kit (Magen, Guangzhou, China). First-strand cDNAs were synthesized using random primers, 1 μg total RNA per 20 μL reaction, and the First Strand cDNA Synthesis Kit (TOYOBO, Osaka, Japan). Quantitative PCR was carried out using the SYBR Green qRT-PCR kit (Vazyme, Nanjing, China) on an Applied Biosystems QuantStudio 6 Flex system (Applied Biosystems, Foster City, CA, USA). Relative expressions of the assayed genes were calculated using the $2^{-\Delta\Delta Ct}$ method. Each treatment has three biological replicates with three technical replicates each. Each experiment was repeated three times. The primers used in this study are listed in the S3 Table.

## Western blot assay

For total protein extraction, *N. benthamiana* and wheat tissue samples were homogenized individually in a lysis buffer containing 60% SDS, 100 mM Tris-HCl (pH 8.8) and 2% β-mercaptoethanol. Protein samples were analyzed in SDS-PAGE gels through electrophoresis, and then transferred onto nitrocellulose membranes. The blots were incubated in a blocking buffer (5% skim milk and 0.05% Tween 20 in 1×TBS) for 60 min followed by detection using specific primary antibodies and then an HRP-conjugated anti-mouse or anti-rabbit secondary antibody (TransGen Biotech, Beijing, China). Detection signal was visualized using an Amersham Imager 680 machine (GE Healthcare BioSciences, Pittsburgh, PA, USA).

## Identification of phosphorylation site in CRP through LC-MS/MS

CRP-GFP fusion was transiently expressed in wheat protoplasts, and then enriched through immunoprecipitation (IP). The enriched CRP-GFP fusion was detected through western blot using an anti-GFP antibody or through separation in SDS-PAGE gels through electrophoresis followed by Coomassie Brilliant Blue staining. The CRP-GFP band was cut from the gel and digested overnight at 37˚C with trypsin followed by LC-MS/MS.

## Yeast two hybrid (YTH) assay

To investigate the interaction between CRP and TaSAPK7, we co-transformed yeast cells with pADTaSAPK7 and pBD-CRP. To investigate the interaction between CRP and TaUBA2C, we constructed plasmid pAD-TaUBA2C, pAD-TaUBA2C$^{D1}$, pAD-TaUBA2C$^{D2}$, and BD-CRP, BD-CRP$^{S162D}$, BD-CRP$^{S165D}$, BD-CRP$^{S162/165A}$, and BD-CRP$^{S162/165D}$. These plasmids were co-transformed into yeast cells, as indicated in the figure legends. The co-transformed cells were grown on the SD/-Leu/-Trp medium for 72 h and then on the SD/-Trp/-Leu/-His/-Ade medium for 3–5 days. Yeast cells co-transformed with AD-T+BD-Lam or AD-T+BD-53 were used as controls.

## Bimolecular fluorescence complementation (BiFC) assay

To further determine the interaction between CRP and TaSAPK7, we co-expressed CRP-nYFP and TaSAPK7-cYFP or NbSRK-cYFP in leaves of *N. benthamiana* plants through agroinfiltration. At 60 h post agroinfiltration, the infiltrated leaves were harvested and examined under a confocal microscope. CRP-nYFP + GUS-cYFP and GUS-nYFP + TaSAPK7-cYFP were used as controls.

### Firefly luciferase complementation imaging (LCI) assay

The coding sequence of CRP was amplified and inserted into the pCambia1300-nLuc to produce pCRP-nLuc. We also cloned the coding sequence of NbSRK or TaSAPK7 into pCambia1300-cLuc to produce pcLuc-NbSRK or pcLuc-TaSAPK7. These plasmids were individually transformed into Agrobacterium cells. After culturing and induction, Agrobacterium culture carrying pCRP-nLuc was mixed with an equal amount of Agrobacterium culture carrying pcLuc-NbSRK or pcLuc-TaSAPK7. The mixed Agrobacterium cultures were then infiltrated individually into the leaves of *N. benthamiana* plants. Three days later, the same leaves were infiltrated again with a 0.2 mM luciferin (Thermo Scientific, USA) solution followed by the detection of luciferase activity using a low-light cooled CCD imaging apparatus (NightOWL II LB983 with indiGO software). cLuc-TaSAPK7+nLuc and cLuc+CRP-nLuc were used as negative controls, cLuc-SAR+sGF-nLuc were used as positive controls.

### Microscale thermophoresis assay (MST)

The interaction between TaUBA2C and CRP was also determined using Monolith NT.115 kit as instructed (NanoTemper Technologies, Munich, Germany). Briefly, the purified TaUBA2C was labeled using the Monolith NT protein labeling kit as instructed by the manufacturer and mixed with purified CRP, CRP$^{S162/165A}$ or CRP$^{S162/165D}$. After incubation, the mixed protein samples were loaded into the capillaries and analyzed using the NanoTemper analytical software to determine the equilibrium dissociation constant (KD).

### Pull-down assay

To confirm the interaction between TaUBA2C and CRP, CRP$^{S162/165A}$ or CRP$^{S162/165D}$, we mixed purified GST or TaUBA2C-GST with CRP-His, CRP$^{S162/165A}$-His or CRP$^{S162/165D}$-His in a 1×TBS buffer containing 1mM PMSF. After 1 h incubation at 4˚C, 20 μL of immobilized glutathione beads was added into each sample followed by another 1 h incubation at 4˚C. After three rinses in 1×TBS buffer containing 0.05% Tween 20, 5× loading buffer was added to the beads of each sample. The samples were individually boiled for 10 min and then analyzed through western blot assays using an anti-His antibody (TransGen Biotech, Beijing, China).

### *In vitro* phosphorylation assay

Purified TaSAPK7-GST (0.5μg) and CRP-His (0.5μg) were mixed, and incubated for 30 min at 30˚C in 1× kinase reaction buffer containing 5 mM DTT, 10 mM MgCl$_2$, 20 mM Tris–HCl (pH 7.5), 100 μM ATP, and 4 μl of [γ-$^{32}$P]ATP. After incubation, 5× SDS loading buffer was added into each reaction, boiled for 10 min, and then analyzed in 12% SDS–PAGE gels. The radioactive signal was visualized using an Amersham Imager 680 machine (GE Healthcare BioSciences, Pittsburgh, PA, USA).

### Virus-induced gene silencing (VIGS)

To silence *NbSRK* expression in *N. benthamiana*, a 300 bp fragment representing partial sequence of *NbSRK* was inserted into the TRV2 vector [59]. Agrobacterium culture carrying pTRV1 and pTRV2 (referred to as TRV:00), or pTRV1 and pTRV2:NbSRK (TRV:NbSRK) was infiltrated into the leaves of *N. benthamiana* plants, and the infiltrated plants were grown inside a climate growth chamber at 25˚C. After 7 days, the plants were inoculated again with CWMV and then grown at 15˚C. Plants inoculated with TRV:00 were used as controls.

The BSMV-based VIGS was done as previously reported [57]. Briefly, a 300 bp fragment from *TaUBA2C* was cloned into the pBSMVγ vector. Plasmid pBSMVα, pBSMVβ, pBSMVγ,

pBSMVγ:TaUBA2C, and pBSMVγ:TaPDS were individually linearized using specific restriction enzymes. The linearized plasmids were used to produce RNA transcripts using a mMessage mMachine T7 *in vitro* transcription kit as instructed (Ambion, Austin, TX, USA). The resulting RNA transcripts representing BSMVα, pBSMVβ, and pBSMVγ (BSMV:00); BSMVα, pBSMVβ, and pBSMVγ:TaPDS (BSMV:TaPDS); or BSMVα, pBSMVβ, and pBSMVγ:TaU-BA2C (BSMV:TaUBA2C) were mixed, diluted 3/7 (v/v) in 1× FES buffer, and rub-inoculated to the second true leaf of each wheat seedling (10 μL mixed transcripts per leaf). The BSMV:00- and BSMV:TaPDS-inoculated plants were used as controls. The inoculated wheat seedlings were then grown for 24 h in the dark and at 28˚C, and then under a 16 h light/ 8 h dark photoperiod. After 10 days, the plants were inoculated again with CWMV and then grown in a climate house at 15˚C until further analysis.

## Trypan blue staining

The agroinfiltrated leaves were harvested, photographed, and analyzed for cell death through trypan blue staining. The harvested leaves were immersed in a trypan blue solution (60 mL absolute ethanol, 10 ml sterile water, 10 ml lactic acid, 10 ml glycerol, 10 ml water phenol, and 15 mg of Trypan blue) followed by 3–5 min boiling. The stained leaves were de-stained through two or three rinses in a chloral hydrate (2.5 g/ml) solution, and photographed. The relative Trypan blue staining intensity was calculated using IMAGEJ software (http://rsbweb.nih.gov/ij), as previously reported [60].

## DAB staining

To detect $H_2O_2$ accumulation, the agroinfiltrated leaves were harvested and stained with a 3,3′-diaminobenzidine (DAB) solution as described [61]. Briefly, the harvested leaves were immersed overnight in the DAB (1mg/mL) staining solution (Sigma) followed by three to five de-staining in absolute ethanol. As previously reported [60], the relative DAB staining intensity was calculated using IMAGEJ software (http://rsbweb.nih.gov/ij).

## Isolation of total nuclear protein, nucleoplasmic protein, and chromatin-bound protein

The agroinfiltrated leaves were harvested, UV cross-linked, and extracted for nuclei in the Honda buffer as previously reported [62]. For each nuclear pellet, three volumes of 1% SDS, 2 mM EDTA, 10 mM Tris (pH 7.5), and PMSF were added and the sample was vortexed for 1 min followed by 10 min centrifugation at 14,000*g*. The soluble nucleoplasmic protein-containing supernatant was collected first, and the remaining pellet was resuspended again in three volumes of 1% SDS solution, sonicated, and then centrifuged at 14,000*g* for 10 min. The chromatin-bound protein-containing supernatant was then collected. The insoluble nuclear protein was prepared through addition of three volumes of 1% SDS solution to the resulting pellet. A 5× sample buffer was added (1:4, v/v) to each soluble nucleoplasmic fraction, chromatin-bound protein fraction or the insoluble nuclear protein fraction followed by 10 min boiling. The resulting samples were analyzed in 12% SDS–PAGE gels followed by antibody labeling.

## RNA- and DNA-binding activity assay

The RNA and DNA binding activity of TaUBA2C was determined using resins containing polyguanylic, polyuridylic, polycytidylic ribohomopolymers, calf thymus single-stranded or double-stranded DNA, respectively (Sigma-Aldrich). These resins were incubated with

purified TaUBA2C-GST (approximately 100 ng per reaction) in 400 μL KHN buffer [0.01% Nonidet P-40, 20 mM HEPES (pH 7.9), 150 mM KCl, and 1× PMSF] for 1h at 4˚C. The bound protein was eluted from the resins using a 5× SDS buffer after 10 min incubation at 80˚C. An aliquot (15 μL) was taken from each sample and analyzed through western blot assay using an anti-GST antibody (TransGen Biotech, Beijing, China).

## RIP-RT-qPCR Assays

Two-week-old WT and TaUBA2C-His transgenic wheat seedlings were collected for cross linking to detect the association of TaUBA2C with *TaNPR1*, *TaPR1* and *TaRBOHD* pre-mRNA using the RIP assay as previously reported [62,63]. Each sample was cross linked in 1% (v/v) formaldehyde, and then two grams of fixed sample was ground and re-suspended in 2mL of Honda buffer (5 mM of DTT, 10 mM of MgCl$_2$, 20 mM of HEPES KOH, 0.44 M of Suc, 0.5% [v/v] Triton X-100, 1.25% [w/v] Ficoll, 2.5% [w/v] dextran, 1 mM of PMSF, 1× cocktail protease inhibitor and 8 U/mL of RNase inhibitor) to lyse the plant cells. The nucleoprotein was treated in nuclear lysis buffer (1% [w/v] SDS, 1× cocktail protease inhibitor, 1 mM PMSF, 10 mM EDTA, 50 mM Tris-HCl and 160 U/mL RNase inhibitor) for sonicating. The sonication product was treated with 20 U/mL DNase I, 200 μL lysate was retained using as the input sample. The remainder was used for immunoprecipitation and incubated with His antibodies at 4˚C overnight. The immunoprecipitation elution buffer was used to wash the immunoprecipitated complexes. The input mRNAs and immunoprecipitated mRNAs were extracted with the HiPure plant RNA mini kit (Magen, Guangzhou, China). The associated pre-mRNA and input pre-mRNA were treated with DNase I and reversed the RNA-protein cross-link. 1 μg RNA from each sample were reverse transcribed as described above, and then qRT-PCR was performed to determine the relative enrichment of each gene. RIP signals were presented as the percentage of immunoprecipitated *TaNPR1*, *TaPR1* and *TaRBOHD* pre-mRNA relative to input *TaNPR1*, *TaPR1* and *TaRBOHD* pre-mRNA.

## Supporting information

**S1 Fig. Phosphorylation of CRP at S162 and S165 does not affect its VSR activity. A.** sGFP was co-expressed with CRP, CRP$^{S162/165A}$ (a non-phosphorylatable CRP), CRP$^{S162/165D}$ (a phosphorylatable CRP), P19, or an empty vector (EV) in the leaves of 16c transgenic *N. benthamiana* plants. The infiltrated leaves were photographed at 3 (left) and 6 dpi (right) under a long-wave UV light. **B.** Detection of GFP in the infiltrated leaf tissues through western blot analysis at 3 (left) and 6 dpi (right) using an anti-GFP or an anti-HA antibody. The CBB-stained gel is used to show sample loadings.
(TIF)

**S2 Fig. Alignment using three TaSAPK7 sequences. A.** Alignment using three wheat SAPK7 sequences. The conserved amino acid (aa) residues are shown in black, while the variable aa residues are shown in blue. **B.** Maximum likelihood tree of SnRK2 proteins based on amino acid sequences of kinase domain. Bootstrap values were estimated based on 1,000 replications. **C.** Alignment using NbSRK and TaSAPK7 (TraesCS2A02G303900.1) sequences.
(TIF)

**S3 Fig. Effect of S156 and T159 mutant on CWMV infection. A.** LC-MS/MS analysis of the phosphorylation status of CRP$^{S162/165D}$ mutant. The underlined CRP$^{S162/165D}$ amino acid sequence was identified in this study through LC-MS/MS, and the secondary phosphorylation sites in this protein are shown in blue. **B.** Systemic mosaic symptoms in the CWMV-, CWMV$^{S156/T159A}$- or CWMV$^{S156/T159D}$-infected *N. benthamiana* plants. Photographs were

taken at 21 dpi. Scale bar = 5 cm (upper panel), Scale bar = 2 cm (lower panel). **C.** Relative expression level of CWMV CP in the assayed *N. benthamiana* plants, determined through qRT-PCR using CWMV CP gene specific primers. The data presented are the means ± standard deviations (SD), calculated using the Student's *t*-test. Each treatment had three biological replicates. n.s, no significant difference. **D.** Accumulation of CWMV CP in the assayed *N. benthamiana* leaf samples was determined through western blot analysis using a CWMV CP specific antibody. The CBB-stained gel is used to show sample loadings.
(TIF)

**S4 Fig. Analyses of *NbSRK* mRNA and protein expressions in plants. A.** Relative expression of *NbSRK* mRNA in the TRV:00+CWMV-, TRV:NbSRK+CWMV-, TRV:NbSRK+-CWMV$^{S162/165A}$- or TRV:NbSRK+CWMV$^{S162/165D}$-inoculated *N. benthamiana* plants was determined through qRT-PCR. The data presented are the means ± SD, determined using the Student's *t*-test. Each treatment had three biological replicates. *, *P* <0.05. **B.** Detection of NbSRK protein accumulation in the *N. benthamiana* leaves co-expressing EV+CWMV, NbSRK+CWMV, NbSRK+CWMV$^{S162/165A}$ or NbSRK+CWMV$^{S162/165D}$ through western blot analysis using an anti-His antibody.
(TIF)

**S5 Fig. Alignment using three TaUBA2C sequences. A.** Alignment using three TaUBA2C sequences. The conserved aa residues are shown in black, while the variable aa residues are shown in blue.
(TIF)

**S6 Fig. Analyses of *TaUBA2C* mRNA and protein expressions in wheat plants. A.** Relative expression levels of *TaUBA2C* mRNA in the BSMV:00+CWMV- or BSMV:TaUBA2C +CWMV-inoculated wheat plants were determined through qRT-PCR at 10 days post BSMV inoculation. The data presented are the means ± SD, determined using the Student's *t*-test. Each treatment had three biological replicates. *, *P* <0.05. **B.** The expression levels of TaUBA2C in eight transgenic wheat lines were determined through western blot analysis using an anti-His antibody.
(TIF)

**S7 Fig. Detection of TaUBA2C, CRP and CRP mutant expressions. A** and **B.** Western blot analyses of TaUBA2C, CRP, CRP$^{S162/165A}$ and CRP$^{S162/165D}$ expressions in plants using an anti-Flag or an anti-HA antibody. **C.** Western blot analysis of TaUBA2C and CRP$^{S162/165D}$ expressions in *N. benthamiana* leaves co-inoculated with TaUBA2C and different concentrations of CRP$^{S162/165D}$ using an anti-Flag or an anti-HA antibody. The CBB-stained gel is used to show sample loadings.
(TIF)

**S8 Fig. Subcellular distribution of CRP and its mutants. A.** Subcellular localization patterns of CRP-GFP, CRP$^{S162/165A}$-GFP and CRP$^{S162/165D}$-GFP in *N. benthamiana* leaf epidermal cells. These proteins were expressed individually in the leaves of the H2B-RFP transgenic *N. benthamiana* plants. Confocal images were taken at 60 hpi. Scale bar = 50 μm. **B.** Western blot analyses of subcellular localizations of the co-expressed TaUBA2C-cYFP and CRP-nYFP, TaUBA2C-cYFP and CRP$^{S162/165D}$-nYFP using an anti-Flag or an anti-HA antibody. Histone H3 protein was used as a nuclear protein marker.
(TIF)

**S9 Fig. BiFC analysis with TaUBA2C$^{ΔNLS}$ and CRP or its mutants.** BiFC analysis with TaU-BA2C$^{ΔNLS}$-cYFP and CRP-nYFP, CRP$^{S162/165D}$-nYFP or GUS-cYFP in H2B-RFP transgenic

*N. benthamiana* leaves. TaUBA2C$^{\Delta NLS}$: nuclear localization signal deletion mutant of TaUBA2C. Confocal images were taken at 60 hpi. Scale bar = 50 μm.
(TIF)

**S10 Fig. Symptoms are classified based on the degree of yellowing and curling of the leaves infected with CWMV.**
(TIF)

**S1 Table. The complete phospho-peptide sequences from the mass spectrum assay.**
(DOCX)

**S2 Table. The total of 15 positive clones obtained from the Y2H screening with the viral CRP as bait.**
(DOCX)

**S3 Table. Primers used in this study.**
(DOCX)

## Acknowledgments

We thank Zhensheng Kang (Northwest Agricultural and Forestry University, Yangling, Shaanxi Province, China) for providing the BSMV-based VIGS vector. We thank Xinshun Ding for critically reading and improving the manuscript.

## Author Contributions

**Conceptualization:** Jian Yang.

**Formal analysis:** Jian Yang.

**Funding acquisition:** Jian Yang, Jianping Chen.

**Investigation:** Juan Li, Huimin Feng, Shuang Liu, Peng Liu, Xuan Chen, Jin Yang, Jian Yang.

**Methodology:** Juan Li, Huimin Feng, Shuang Liu, Peng Liu, Xuan Chen, Jin Yang, Long He, Jian Yang.

**Project administration:** Jian Yang, Jianping Chen.

**Supervision:** Jian Yang, Jianping Chen.

**Writing – original draft:** Juan Li, Jian Yang.

**Writing – review & editing:** Jian Yang, Jianping Chen.

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
