## [Decision Letter · Decision Letter 0]

18 Feb 2022

Dear Dr Yang,

Thank you very much for submitting your manuscript "Phosphorylated viral protein evades plant immunity through interfering the function of RNA-binding protein" for consideration at PLOS Pathogens. As with all papers reviewed by the journal, your manuscript was reviewed by members of the editorial board and by several independent reviewers. The reviewers appreciated the attention to an important topic. Based on the reviews, we are likely to accept this manuscript for publication, providing that you modify the manuscript according to the review recommendations.

Sincerely,

Savithramma P. Dinesh-Kumar

Associate Editor

PLOS Pathogens

Shou-Wei Ding

Section Editor

PLOS Pathogens

Kasturi Haldar

Editor-in-Chief

PLOS Pathogens

orcid.org/0000-0001-5065-158X

Michael Malim

Editor-in-Chief

PLOS Pathogens

orcid.org/0000-0002-7699-2064

Reviewer Comments (if any, and for reference):

Reviewer's Responses to Questions

**Part I - Summary**

Reviewer #1: In the revision, the authors have answered most of my questions and the manuscript has been improved.

Reviewer #2: summary

The manuscript PPATHOGENS-D-22-00083 is an intensively revised version of an earlier manuscript supplemented by extensive additional experiments. In addition, the authors submitted a detailed and satisfactory letter of reply to the referee's comments on the previous manuscript.

As previously noted, the present work addresses the important and hitherto understudied question of how viral-plant interactions are regulated by PTMs of viral proteins. By identifying and functionally characterizing phosphorylation sites in the silencing suppressor CRP of CWMV, as well as identifying a host kinase that phosphorylates CRP and the CRP-interacting TaUBA2C, the authors made important contributions to understanding the interplay between plant defense and viral suppression of defense done.

Although the manuscript has been largely revised satisfactorily, there are still a few small comments to make

**Part II – Major Issues: Key Experiments Required for Acceptance**

Reviewer #1: (No Response)

Reviewer #2: (No Response)

**Part III – Minor Issues: Editorial and Data Presentation Modifications**

Reviewer #1: (No Response)

Reviewer #2: 55 Explain BSMV abbreviation

57 For geminiviruses, there are extensive review articles on PTMS that regulate the functions of viral proteins, including SnRK1, possibly cite them here (W. Shen and L. Hanley-Bowdoin Curr Opin Virol 2021 Vol. 47 Pages 18-24; R. M. Teixeira, M. A. Ferreira, G. A. S. Raimundo and E. P. B. Fontes Microorganisms 2021 Vol. 9 Issue 4; L. Hanley-Bowdoin, E. R. Bejarano, D. Robertson and S. Mansoor Nat Rev Microbiol 2013 Vol. 11 Issue 11 Pages 777-88)

134 remove “include that”?

151 at this point of the manuscript the conclusion is to early, better “suggest that S162 and S165 phosphorylation sites are important”

196 remove “that”

PLOS authors have the option to publish the peer review history of their article (what does this mean?). If published, this will include your full peer review and any attached files.

Reviewer #1: No

Reviewer #2: No

Figure Files:

Data Requirements:

Reproducibility:

References:

---

## [Editor Report · Decision Letter 1]

1 Mar 2022

Dear Dr Yang,

We are pleased to inform you that your manuscript 'Phosphorylated viral protein evades plant immunity through interfering the function of RNA-binding protein' has been provisionally accepted for publication in PLOS Pathogens.

Best regards,

Savithramma P. Dinesh-Kumar

Associate Editor

PLOS Pathogens

Shou-Wei Ding

Section Editor

PLOS Pathogens

Kasturi Haldar

Editor-in-Chief

PLOS Pathogens

orcid.org/0000-0001-5065-158X

Michael Malim

Editor-in-Chief

PLOS Pathogens

orcid.org/0000-0002-7699-2064
---

## [Editor Report · Acceptance letter]

12 Mar 2022

Dear Dr Yang,

We are delighted to inform you that your manuscript, "Phosphorylated viral protein evades plant immunity through interfering the function of RNA-binding protein," has been formally accepted for publication in PLOS Pathogens.

Best regards,

Kasturi Haldar

Editor-in-Chief

PLOS Pathogens

orcid.org/0000-0001-5065-158X

Michael Malim

Editor-in-Chief

PLOS Pathogens

orcid.org/0000-0002-7699-2064